## OPEN

# Structural and functional characterization of the intracellular filament-forming nitrite oxidoreductase multiprotein complex

Tadeo Moreno Chicano[1,8], Lea Dietrich[2,8], Naomi M. de Almeida[3,8], Mohd. Akram[1], Elisabeth Hartmann[1], Franziska Leidreiter [1], Daniel Leopoldus [1], Melanie Mueller[1], Ricardo Sánchez[2,4], Guylaine H. L. Nuijten[3], Joachim Reimann[3], Kerstin-Anikó Seifert[1], Ilme Schlichting [1], Laura van Niftrik [3], Mike S. M. Jetten[3], Andreas Dietl[1 ✉], Boran Kartal[5,6 ✉], Kristian Parey[2,7 ✉] and Thomas R. M. Barends [1 ✉]

Nitrate is an abundant nutrient and electron acceptor throughout Earth's biosphere. Virtually all nitrate in nature is produced by the oxidation of nitrite by the nitrite oxidoreductase (NXR) multiprotein complex. NXR is a crucial enzyme in the global biological nitrogen cycle, and is found in nitrite-oxidizing bacteria (including comammox organisms), which generate the bulk of the nitrate in the environment, and in anaerobic ammonium-oxidizing (anammox) bacteria which produce half of the dinitrogen gas in our atmosphere. However, despite its central role in biology and decades of intense study, no structural information on NXR is available. Here, we present a structural and biochemical analysis of the NXR from the anammox bacterium *Kuenenia stuttgartiensis*, integrating X-ray crystallography, cryo-electron tomography, helical reconstruction cryo-electron microscopy, interaction and reconstitution studies and enzyme kinetics. We find that NXR catalyses both nitrite oxidation and nitrate reduction, and show that in the cell, NXR is arranged in tubules several hundred nanometres long. We reveal the tubule architecture and show that tubule formation is induced by a previously unidentified, haem-containing subunit, NXR-T. The results also reveal unexpected features in the active site of the enzyme, an unusual cofactor coordination in the protein's electron transport chain, and elucidate the electron transfer pathways within the complex.

Microorganisms turn over several trillion tons of nitrogen every year, and so control the bioavailability of this essential element[1]. In marine and terrestrial ecosystems, nitrate is one of the most abundant reactive nitrogen species and is used for assimilation or respiration by many groups of organisms[1]. In nature, nitrate is produced almost exclusively by the oxidation of nitrite, which keeps nitrogen bioavailable. Moreover, nitrite oxidation is an important step in wastewater treatment processes. Nitrite oxidation to nitrate is carried out by the nitrite oxidoreductase (NXR) complex, but despite its vital importance in the nitrogen cycle, NXR is one of this cycle's least understood enzymes[2]. Aerobic nitrite-oxidizing bacteria (including comammox organisms), which generate the bulk of the nitrate in the environment, and anaerobic ammonium-oxidizing (anammox) bacteria, which are responsible for producing half of the dinitrogen ($N_2$) gas in the atmosphere, are the two main NXR-encoding clades of microorganisms[1,2].

NXRs are genetically diverse[3–7]. The phylogenetically distinct anammox NXRs are most closely related to the membrane-bound or soluble NXRs from nitrite-oxidizing *Nitrospira* (including comammox *Nitrospira*)[3,6], *Nitrospina*[5] and more distantly to *Nitrotoga*[6], which all have NXRs that are suggested to be periplasmically oriented. This clade is also phylogenetically related to the *Beggiatoa* and *Hydrogenobaculum* nitrate reductases[5]. The NXRs of *Nitrolancea*, phototrophic *Thiocapsa* KS1 (ref. [7]) as well as the membrane-bound, cytoplasmically oriented NXRs of the '*Nitrobacter* type' (such as *Nitrobacter* and *Nitrococcus*) form another distinct group, which is more closely related to the nitrate reductases from, for example, *Escherichia coli*[3–5].

In aerobic nitrite oxidizers, NXRs are typically membrane bound (with some possible exceptions[6,8]), and the electrons that are generated by nitrite oxidation are ultimately donated to molecular oxygen to yield energy for growth[2]. In anammox bacteria, however, NXRs are associated with tubule structures of hitherto unknown architecture and function inside the 'anammoxosome'[9], the bacterial organelle in which the anammox substrates nitrite and ammonium are converted to $N_2$ and nitrate[10]. Recent results indicate that the electrons generated by anammox NXR are used for the reduction of nitrite to nitric oxide (NO)[11], a central intermediate in the anammox process[12,13]. NXRs have been only partially purified from nitrite-oxidizing bacteria, and their structure and molecular mechanism remain unknown[2].

[1]Department of Biomolecular Mechanisms, Max Planck Institute for Medical Research, Heidelberg, Germany. [2]Department of Structural Biology, Max Planck Institute of Biophysics, Frankfurt am Main, Germany. [3]Department of Microbiology, Radboud University, Nijmegen, the Netherlands. [4]Buchmann Institute for Molecular Life Sciences, Goethe University of Frankfurt am Main, Frankfurt am Main, Germany. [5]Microbial Physiology Group, Max Planck Institute for Marine Microbiology, Bremen, Germany. [6]Department of Life Science and Chemistry, Jacobs University Bremen, Bremen, Germany. [7]Present address: Department of Structural Biology, University of Osnabrück, Osnabrück, Germany. [8]These authors contributed equally: Tadeo Moreno Chicano, Lea Dietrich, Naomi M. de Almeida. ✉e-mail: Andreas.Dietl@mpimf-heidelberg.mpg.de; bkartal@mpi-bremen.de; Kristian.Parey@biophys.mpg.de; Thomas.Barends@mpimf-heidelberg.mpg.de

In contrast to the membrane-bound respiratory nitrate reductase (NAR), which lacks nitrite-oxidizing activity, physiological studies have shown that NXRs can catalyse both nitrite oxidation and nitrate reduction[1,2,14–16]. However, how NXRs control the direction of their electron flow remains unclear. Moreover, how anammox NXRs transfer the electrons harvested from nitrite oxidation to their redox partners, and why and how these enzymes form tubule structures in anammox bacteria is not understood. Studying anammox bacteria is extremely challenging, given the lack of genetic tools for them and the fact that the multicofactor metalloproteins driving their metabolism cannot be expressed heterologously. Thus, virtually all biochemical anammox research must be performed on wild-type proteins purified directly from anammox biomass, and mutational studies cannot be performed.

Here, we present a multiscale structural and biochemical analysis of the NXR from the anammox bacterium *Kuenenia stuttgartiensis*, integrating X-ray crystallography, cryo-electron tomography and helical reconstruction, interaction and reconstitution studies and enzyme kinetics. We find unexpected features in the active site and an unusual cofactor coordination in the protein's electron transport chain. Moreover, we demonstrate that NXR itself is the main constituent of the anammoxosomal tubule structures and we identify a previously unidentified, unexpected subunit crucial to tubule formation.

In anammox bacteria, NXR is encoded on a gene cluster conserved throughout the known anammox genera[13,17] (Fig. 1a). In *K. stuttgartiensis*, the *nxr* gene cluster comprises 15 genes (*kustd1699–kustd1713*), which encode several proteins homologous to subunits of membrane-bound NAR. Kustd1700 is similar to the catalytic NarG subunit of NAR (Extended Data Fig. 1, Supplemental Information). Kustd1700 belongs to the DMSO reductase superfamily of molybdopterin-containing enzymes[18–21] and as such is predicted to contain two molybdopterin guanosine dinucleotide cofactors as well as a [4Fe–4S] iron–sulfur cluster (FS0) close to its active site. The Kustd1703 subunit is homologous to NarH and is predicted to contain three additional [4Fe–4S] clusters and a single [3Fe–4S] iron–sulfur cluster (FS1–FS4). Together, these clusters form a chain that allows electron transport to or from the active site[15,22]. In general, NXRs are believed to also contain a haem-containing C-subunit, which in nitrite-oxidizing bacteria is expected to be a membrane protein[15,23–25], with some possible exceptions[5,6,8]. In the anammox bacterium *K. stuttgartiensis*, however, the NXR cluster encodes kustd1704, which is homologous (24% identity) to the C-subunit of ethylbenzene dehydrogenase (EBDH) from *Aromatoleum aromaticum*[26], another member of the DMSO reductase family. The EBDH C-subunit is not membrane bound but does contain a single, *b*-type haem. It was thus initially hypothesized that the catalytic component of *K. stuttgartiensis* NXR would not be a membrane-bound complex[13,27]. Therefore, we isolated NXR from the soluble fraction of *K. stuttgartiensis* cell lysate. As expected, the purified protein comprised the gene products of *kustd1700* (NXR-A), *kustd1703* (NXR-B) and *kustd1704* (NXR-C). This preparation of *K. stuttgartiensis* NXR (KsNXR-ABC) displayed the typical features of a high redox potential *b*-type haem in ultraviolet–visible (UV–Vis) spectroscopy (Extended Data Fig. 2a,b and Supplemental Information), and the presence of a pterin prosthetic group was suggested by fluorescence spectroscopy on permanganate-oxidized samples (Extended Data Fig. 2c and Supplemental Information). KsNXR-ABC stoichiometrically oxidized nitrite to nitrate (Fig. 1b), with a $V_{max}$ of $61 \pm 4$ nmol min$^{-1}$ mg protein$^{-1}$ and an apparent $K_m$ of $22 \pm 5\,\mu M$ with ferricyanide as the artificial electron acceptor. Nitrate was reduced in the presence of the artificial electron donor methyl viologen (VM) ($E_0' = -0.45\,V$ at pH 7) with a $V_{max}$ of $1.5 \pm 0.05\,\mu mol\,min^{-1}\,mg\,protein^{-1}$ and an apparent $K_m$ of $14 \pm 1\,\mu M$, demonstrating that KsNXR-ABC can indeed act as a bidirectional enzyme.

We then determined the 3.0-Å resolution crystal structure of KsNXR-ABC, which displays heterotrimers shaped like a curved funnel (Fig. 1c,d, Supplemental Information, Extended Data Fig. 3a and Supplementary Table 1), reminiscent of NAR[28,29] and EBDH[26]. At the top of the A subunit, a narrow, 25-Å long tunnel leads to the active site molybdenum ion. In the active site itself (Fig. 2a), the conserved Asp275A coordinates the molybdenum, although at the present resolution (3.0 Å) a distinction between monodentate and bidentate coordination cannot be made. Opposite the molybdenum, the conserved Asn70A is in a perfect position to coordinate substrate molecules bound to the molybdenum, as proposed for the homologous residue in NAR[29,30]. Typically for a member of the DMSO reductase family, the molybdenum is further coordinated by two molybdopterin guanosine dinucleotide molecules. Strikingly, in both of these, the pyran rings are in the 'open' state (Extended Data Fig. 3b and Supplementary Information). A chain of iron–sulfur clusters connects the active site with the C-subunit haem, starting with the FS0 cluster located at ~7 Å distance (edge-to-edge) from the proximal pterin. This cluster is unusual, being coordinated by three cysteines and Asp71 (Fig. 2a). An Asp in this sequence position was noted previously[3]. By contrast, in the structures of NAR and EBDH, this cluster shows a 3Cys/1His coordination, with the histidine being in the position of Cys67 in KsNXR-A. A multiple sequence alignment of NXRs from various lineages and NARs (Extended Data Fig. 1) reveals that this cluster is likely either 3Cys/1Asp or 3Cys/1His coordinated (depending on the lineage) in both types of enzymes. Coordination by an aspartate or histidine affects the redox potential of a [4Fe–4S] cluster[31], and the apparent conservation of such coordination of FS0 in both NXRs and NARs suggests that tuning the redox potential of this cluster has functional importance. The other FeS clusters are coordinated by cysteines only, which appear to be conserved in NXRs and NARs alike. At the other end of the chain of iron–sulfur clusters, the *b*-type haem in the C-subunit is coordinated by Met155C and Lys305C as in EBDH[26] (Fig. 2b), which likely contributes to its high redox potential.

Whereas NXR is associated with tubule structures in the anammoxosome organelle in vivo, isolated KsNXR-ABC did not readily form such structures in vitro. We therefore performed cryo-electron tomography on *K. stuttgartiensis* cells and obtained a 22-Å resolution map of the anammoxosomal tubules by subtomogram averaging (Fig. 3a–d, Extended Data Fig. 4 and Supplementary Table 2). This revealed that the tubules consist of head-to-tail dimers of NXR-ABC heterotrimers (Fig. 3d) also observed in the asymmetric unit of the crystal structure (Extended Data Fig. 3 and Supplementary Information), held together by an additional density that could not be identified immediately. We next prepared highly pure tubules from *K. stuttgartiensis* biomass. These tubules, which we found can be disassembled by adding the detergent dodecyl maltoside, show nitrite oxidation activity (Extended Data Fig. 5 and Supplementary Information). Proteomic analysis of the purified tubules revealed (next to KsNXR-ABC) the presence of Kustd1705 (Supplementary Information and Extended Data Fig. 5), a monoheme cytochrome *c* encoded on the *nxr* gene cluster in *K. stuttgartiensis* (Fig. 1a), suggesting that Kustd1705 could explain the additional density in the tubule structure. Indeed, when NXR-ABC was incubated with Kustd1705, tubule formation was observed (Fig. 3e,f and Supplementary Information). We therefore propose to name this unique protein (Kustd1705) 'NXR-T', for tubule-inducing NXR subunit.

In cell cultures, NXR-T is expressed at similar levels to the other NXR subunits, also when cells are fed on NO rather than nitrite[11]. Using helical reconstruction cryo-electron microscopy (cryo-EM) on reconstituted tubules, we determined the structure of KsNXR-T at 5.8 Å resolution (Supplementary Information, Extended Data Figs. 6 and 7 and Supplementary Table 3) and its positioning in the tubules. Moreover, we determined the 4.0-Å resolution crystal

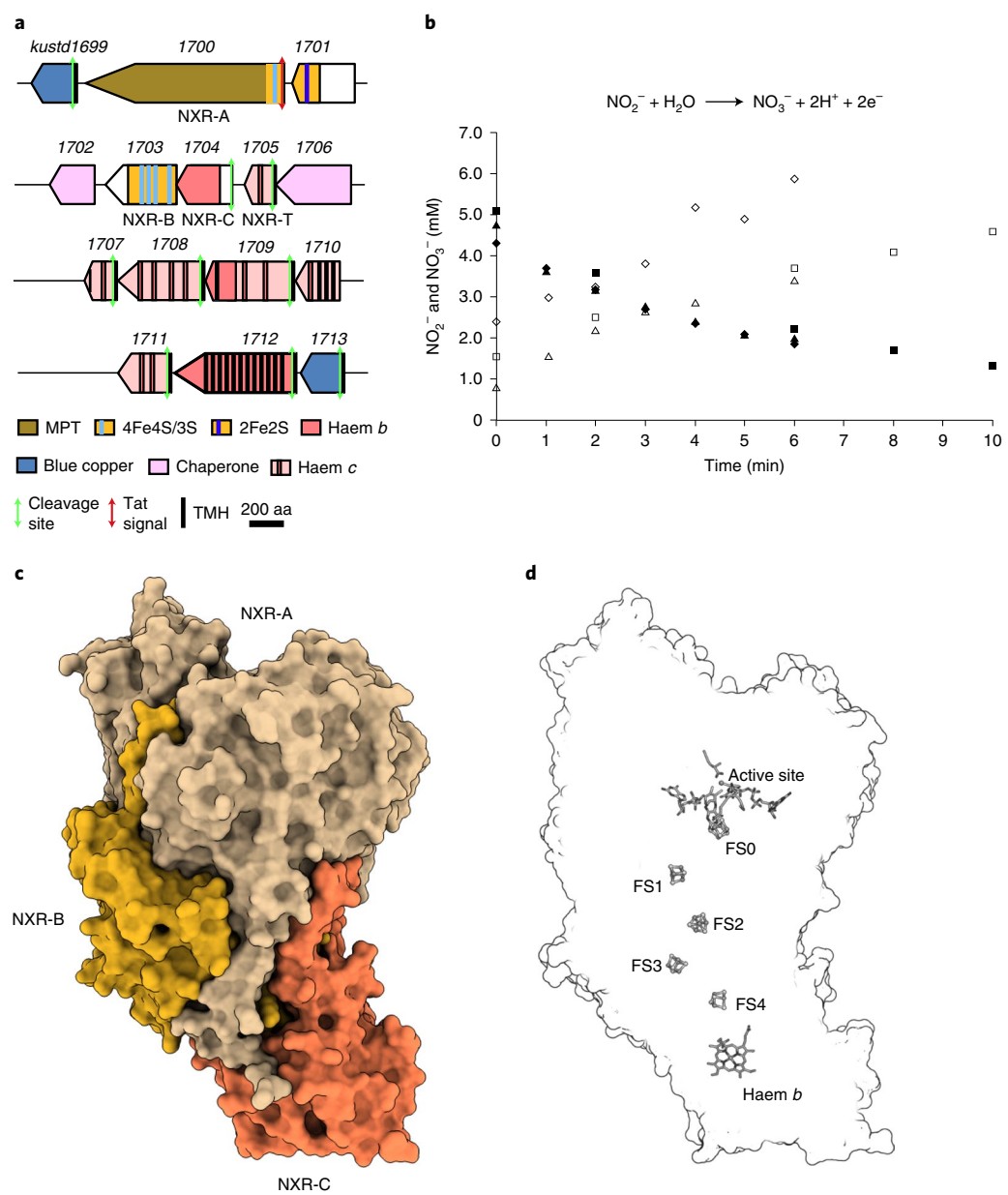

**Fig. 1 | KsNXR enzymatic properties and heterotrimeric structure. a**, Build-up of the NXR operon in *K. stuttgartiensis*. **b**, Nitrite oxidation to nitrate with 1:1 nitrite/nitrate stoichiometry catalysed by KsNXR-ABC. Kinetic traces from three separate experiments are shown; nitrite concentrations are shown as filled symbols (diamonds, triangles and squares), nitrate concentrations as their open equivalents. **c**, KsNXR-ABC crystal structure. A single heterotrimer is depicted. The NXR-A, -B and -C subunits are shown in beige, orange and red, respectively. **d**, Outline of the NXR-ABC heterotrimer with the positions of the cofactors therein. MPT, molybdopterin; aa, amino acids.

structure of the homologue Broful_01488 (43% sequence identity) from the anammox bacterium *Brocadia fulgida* (Supplementary Information and Extended Data Fig. 7) which has a highly similar fold. This allowed construction of a molecular model of the anammoxosomal tubules (Fig. 4). KsNXR-T is a four-helical bundle, occurring as a dimer both in the tubules and in solution (Fig. 4 and Supplementary Information). The haem group of NXR-T is bound close to the C terminus, which itself extends towards the substrate tunnel on the adjacent NXR-A subunit. Spectropotentiometry showed that this haem has a redox potential of +120 mV versus the standard hydrogen electrode (SHE) (Extended Data Fig. 7).

NXR oxidizes nitrite to nitrate in the molybdopterin-containing active site of the A subunit. The resulting electrons are then transferred to the *b*-type haem in the C-subunit via the chain of iron–

sulfur clusters. From here, they cannot travel further through the NXR structure; in the tubules, this haem is too far away (>40 Å) from the *c*-type haem in NXR-T for the electrons to be transferred there efficiently[32,33] (Fig. 4). Moreover, the redox potential of NXR-T (+120 mV) is much lower than that of the nitrite/nitrate redox couple (+420 mV), making it unlikely that the electrons generated by nitrite oxidation are ultimately transferred there. Thus, these electrons are most likely transferred from the *b*-type haem in NXR-C to another binding partner. Interestingly, the NXR-C haem lies behind a positively charged surface groove, which in the tubules lies on the outside of the superstructure (Fig. 4d,f and Extended Data Fig. 8). Because the electrons produced by NXR through nitrite oxidation are used to reduce nitrite to NO[11], these electrons have to be transferred to the NO-generating nitrite reductase of the

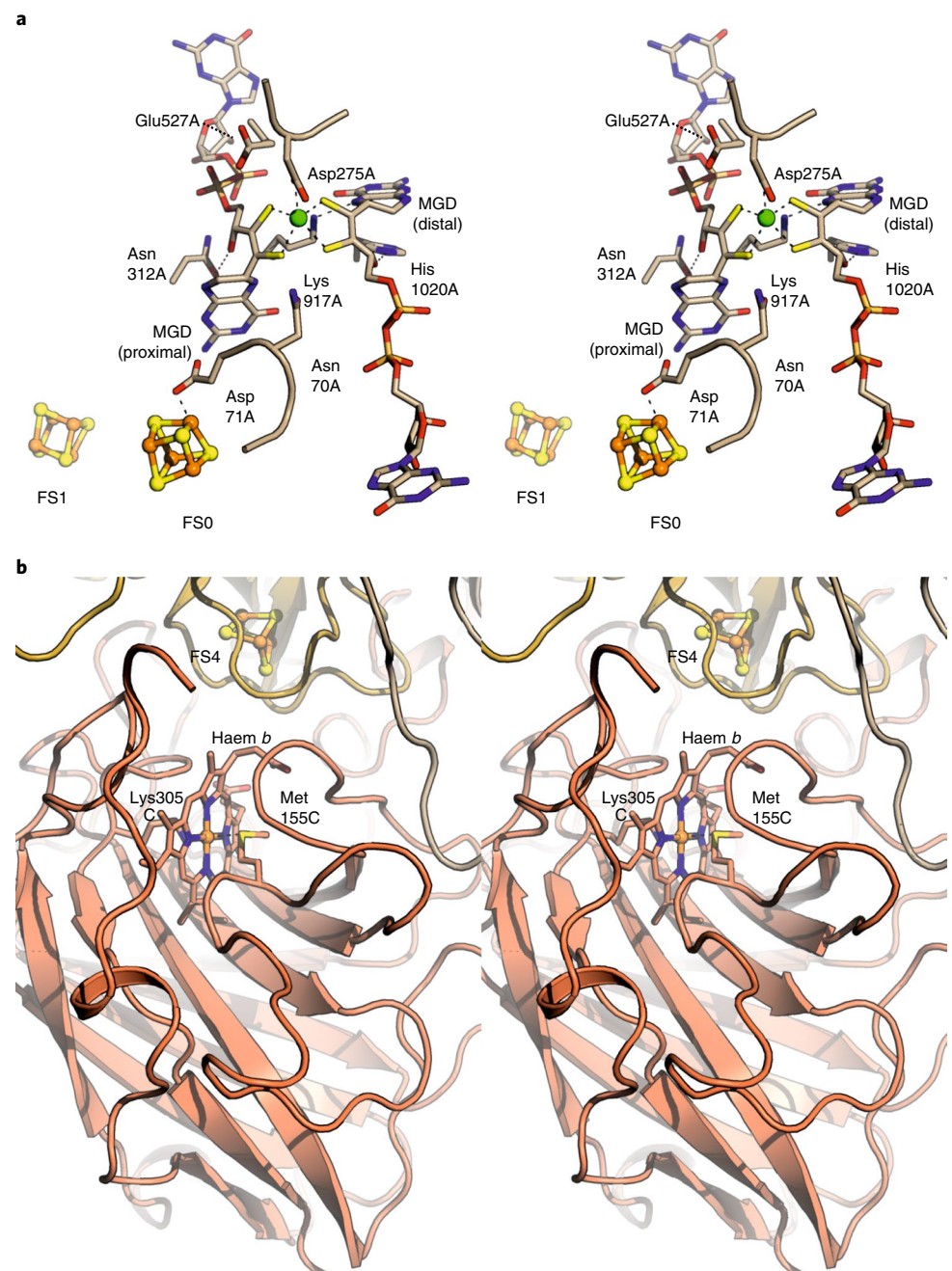

**Fig. 2 | Details of the KsNXR active and haem sites. a**, Stereofigure, showing a close-up of the active site in NXR-A. Active site residues (text and Supplementary Information) are shown as sticks, the FS0 and FS1 iron–sulfur clusters as ball-and-stick models. The molybdenum ion is shown as a green sphere. **b**, Stereofigure of the haem-binding site in NXR-C, close to the interface with NXR-B. The haem and the nearby FS4 iron–sulfur cluster are shown as sticks and a ball-and-stick model, respectively. MGD, molybdopterin guanine dinucleotide.

central anammox metabolism[34–37], either directly or via an electron shuttle as proposed earlier[11]. A redox partner could access this site to accept electrons from the haem cofactor for transfer to the nitrite reductase. Such direct coupling to the nitrite oxidation system explains why in anammox bacteria NXR is localized inside the anammoxosome, rather than on a membrane as in nitrite-oxidizing bacteria, because the NO-producing enzymes in anammox bacteria are soluble, anammoxosomal proteins. Moreover, by having NXR inside the anammoxosome, the protons produced by nitrite oxidation contribute to the proton gradient across the anammoxosomal membrane, analogous to what has been suggested[3] for the periplas-

mically oriented *Nitrospira*-type NXRs, which are phylogenetically closely related to anammox NXR.

However, although localization of NXR inside the anammoxosome rather than on a membrane is rationalized relatively easily, the question of why NXR forms tubules in anammox organisms remains. Interest in enzymes forming intracellular tubules, filaments or similar assemblies has recently surged, and a wide range of functions for the formation of such superstructures is being proposed[38,39] including stabilization, specificity control and recruitment of binding partners. It is also not known at present if the NXR-T subunit identified in this study plays a role beyond tubule formation. Interestingly,

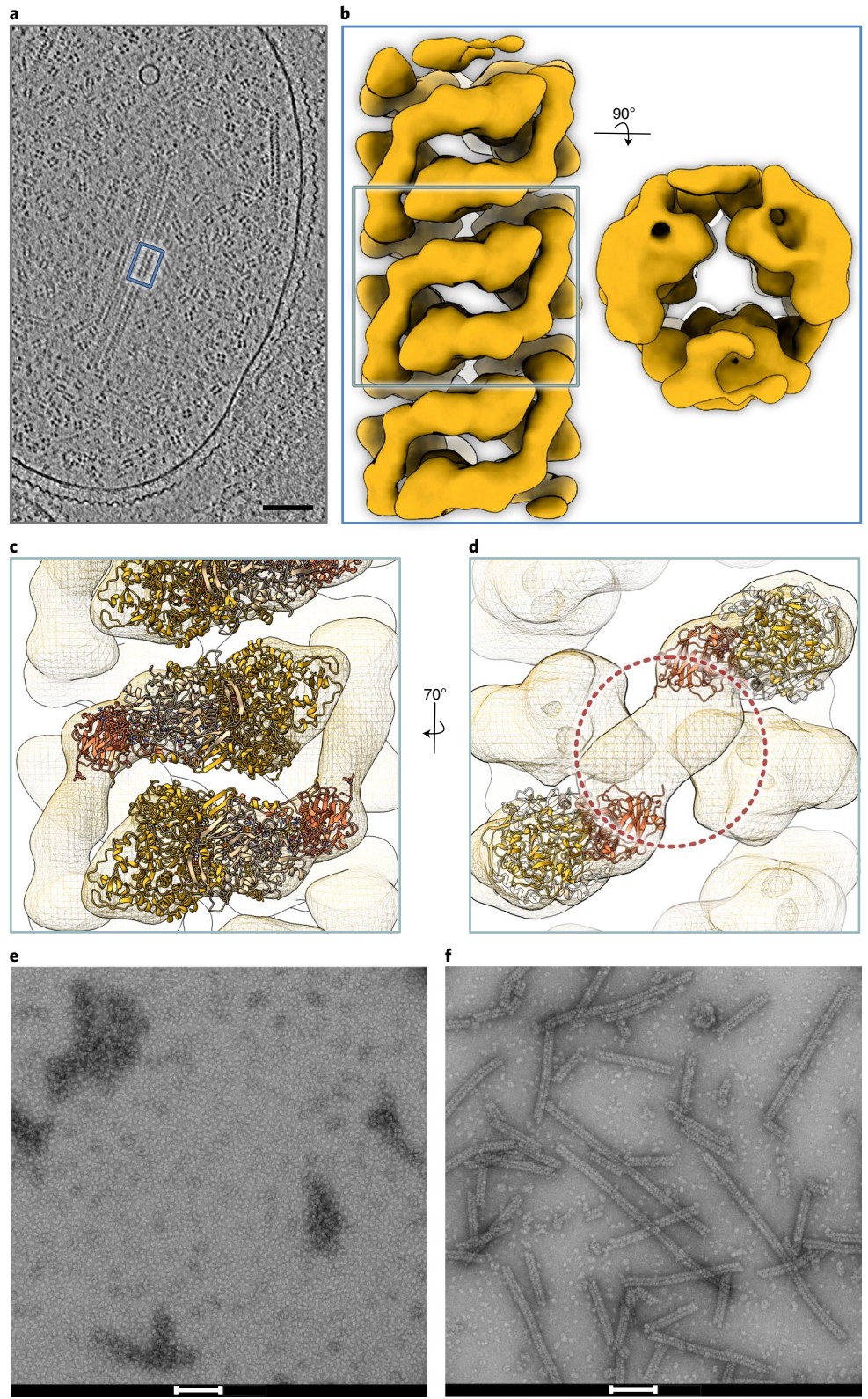

**Fig. 3 | Structure of the anammoxosomal tubules. a**, Slice through a tomogram of a *K. stuttgartiensis* cell. Elongated structures such as those inside the blue box are anammoxosomal tubules; the fourfold symmetrical particles are hydrazine dehydrogenase (HDH) complexes[76]. Scale bar, 100 nm. Shown is a slice from one representative tomogram of 19. **b**, A 22 Å resolution subtomogram average of anammoxosomal tubules. A side and top view of the tubule are shown. **c**, Fit of KsNXR-ABC trimers (cartoon) into the subtomogram averaging map. **d**, The connecting density (circled) is not explained by KsNXR-ABC. **e,f**, Negative-stain electron micrographs of KsNXR-ABC alone (**e**) and after incubation with KsNXR-T (**f**). Scale bars, 100 nm. Shown are representative micrographs out of three independent reconstitution reactions.

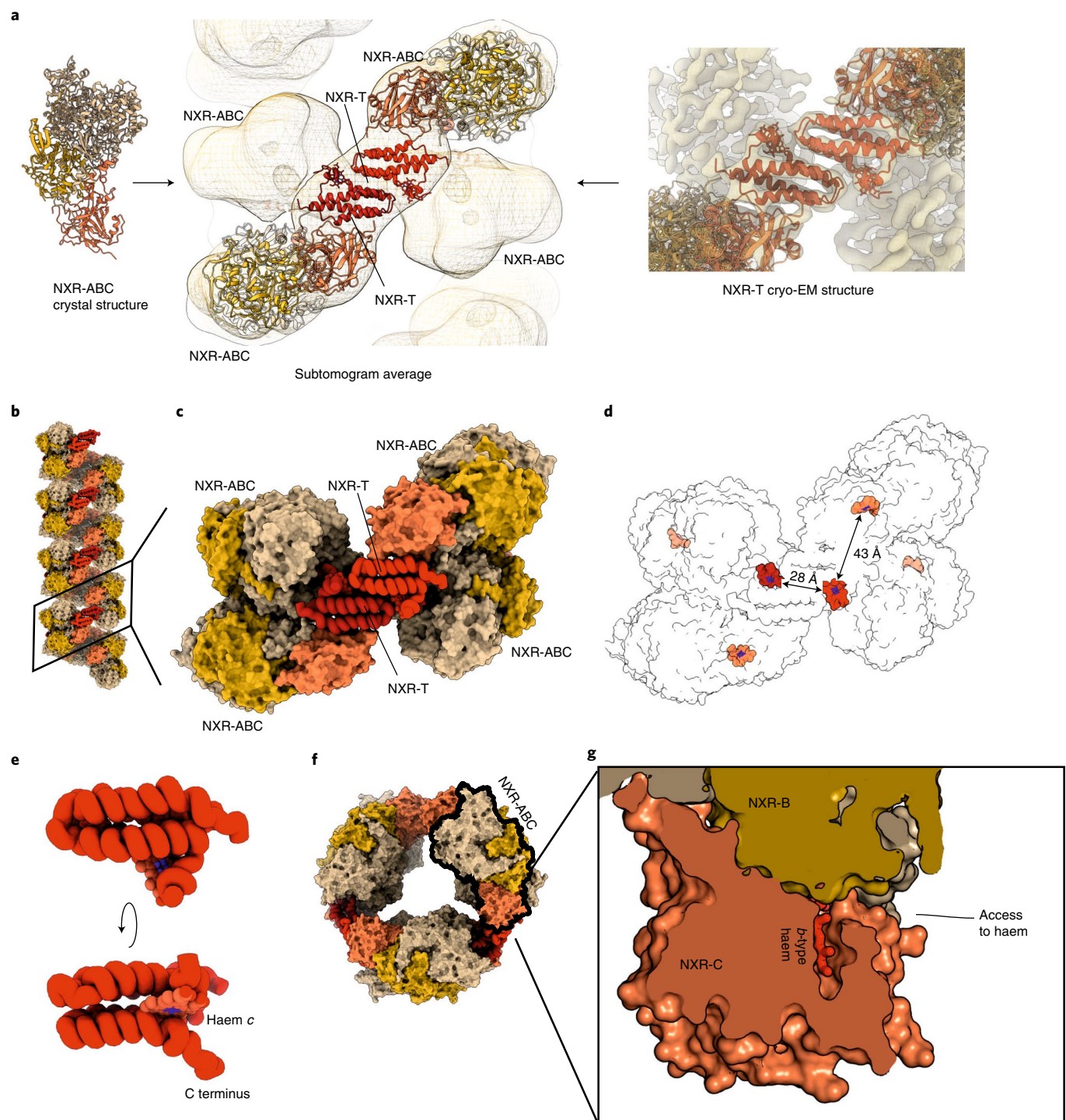

**Fig. 4 | Details of the tubular NXR superstructure. a**, Reconstruction of the tubule structure. The crystal structure of NXR-ABC and the cryo-EM structure of NXR-T were fitted into the subtomogram averaging result of NXR tubules inside the cell. **b**, Reconstructed tubule; the various protein subunits are coloured individually. **c**, Close-up, showing the interactions between NXR-T (red tubes) and NXR-ABC (surface representation). **d**, Distances between the haem cofactors (red) in NXR-T and NXR-ABC. **e**, Fold of KsNXR-T. **f**, View of an NXR tubule down its long axis. A single NXR-ABC trimer is outlined in black. **g**, Slice through an NXR-ABC trimer around the NXR-C haem, which lies behind a cleft in NXR-ABC on the outside of the tubule structure, enabling access to the cofactor.

however, although NXR-A, -B and -C are highly conserved between anammox genera (as are most other anammox metabolic proteins), NXR-T shows much less conservation (Extended Data Fig. 7) and is only part of the NXR gene cluster in *K. stuttgartiensis*, whereas NXR-T homologues are present on distinct genetic loci in the other anammox genera. This is particularly interesting given that the various anammox genera appear to use different types of enzymes for NO generation[10,17]. For instance, *K. stuttgartiensis* was proposed to employ a highly expressed nitrite-reducing octaheme *c*-type cytochrome[10,34,36], whereas the cytochrome *cd*₁-containing NirS

nitrite reductase also encoded in its genome is hardly transcribed[10]. *Scalindua profunda*, on the other hand, contains both a highly abundant NirS and an octaheme *c*-type cytochrome that could perform nitrite reduction, meaning that there are two likely NO-generating enzymes in this organism[40], and in *Jettenia caeni* a copper-based NirK-type nitrite reductase was proposed to be involved in nitrite reduction for NO generation[37]. This variability between genera suggests that NXR-T might play a role in the coupling between the highly conserved NXR-ABC subunits, on the one hand, and the diverse NO-generating enzyme(s) encountered in the various genera, on the other. Possibly, NXR-T mediates interactions between the NXR tubules and either the NO-generating enzyme(s) itself, or a protein shuttling electrons to it.

## Methods

**Purification of KsNXR-ABC.** Chemicals used for purification were obtained from Sigma Aldrich or Merck, unless stated otherwise. HPLC grade chemicals were purchased from Baker. For desalting and concentration of samples, 100 kDa molecular mass cut-off spin-filters were used (Vivaspin 20 or 500, Sartorius Stedim Biotech). Heterotrimeric nitrite oxidoreductase (NXR-ABC) was purified from a *K. stuttgartiensis* enrichment culture (>95%). Cells were cultured as planktonic cells in a 10-l bioreactor[12] and were collected by centrifugation (2 l, optical density ~ 1, 4000*g* for 15 min). The cell pellet was resuspended in 20 ml of 20 mM Tris–HCl buffer pH 8.0, and subsequently disrupted in a French pressure cell at 138 MPa in three passages. Cell debris was removed by a second centrifugation step (4,000*g* for 15 min), and the supernatant was subjected to ultracentrifugation (180,000*g* for 1 h). The resulting pellet was resuspended in 6 ml of 20 mM Tris–HCl buffer pH 8.0, and after incubation with 2% (w/v) *n*-dodecyl-β-D-maltoside (5 r.p.m. for 1 h), ultracentrifugation was repeated and the resulting supernatant was combined with the first supernatant, yielding a dark red cell-free extract.

All fast protein liquid chromatography steps were performed on an ÄKTA purifier system (GE Healthcare) at 4 °C. Depending on the column material and diameter, the fast protein liquid chromatography columns were run at 5 or 2 ml min⁻¹. Protein elution was followed online by the absorption at 280 nm and samples were collected in 2-ml fractions. As a first step, the cell-free extract was loaded onto a 60-ml Q Sepharose XL column (GE Healthcare) and washed with 20 mM Tris–HCl buffer, pH 8.0, until the absorbance at 280 nm was stable below 100 mAU. Nitrate-reducing fractions were eluted with 20 mM Tris–HCl buffer, pH 8.0, containing 200 mM NaCl. The collected fractions were desalted in 10 mM potassium phosphate (KP$_i$) buffer, pH 8.0, and concentrated to a volume of less than 10 ml. The sample was subsequently loaded onto a 30-ml CHT ceramic hydroxyapatite column (Bio-Rad) which was equilibrated with ~75 mM KP$_i$ buffer, pH 8.0. Proteins were eluted with a 30-min linear gradient of 75 to 500 mM KP$_i$ buffer, pH 8.0. After run-off, the only eluting protein as represented by a single peak was collected, desalted in 10 mM KP$_i$ buffer, pH 6.0 and concentrated to 5 ml. The sample was loaded onto a SP Sepharose Fast Flow column (GE Healthcare), equilibrated with 30 mM KP$_i$ buffer, pH 6.0. The run-off was collected, concentrated to <500 µl and loaded onto a 560-ml Sephacryl S-400 HR gel-filtration column (GE Healthcare) equilibrated with 50 mM KP$_i$ buffer, pH 7.25, 150 mM NaCl. Fractions containing only KsNXR-ABC as judged by UV–Vis scans and denaturing gels were pooled, concentrated into ~3.5 mg ml⁻¹ and stored at 4 °C until further use. The identity and purity of NXR were routinely determined as described below.

**Detection of molybdopterin.** Purified KsNXR-ABC (100 µg) was diluted in 10 mM KP$_i$ buffer, pH 7.0, 0.1 M NaOH, 10 µM potassium permanganate and heated for 2 h at 80 °C. Ethanol (0.2 ml) was added to stop the reaction and reduce the excess permanganate[41]. The resulting manganese oxide precipitate was removed by centrifugation in a bench-top centrifuge. As positive control, a few flocks of the oxidized derivate pterine-6-carboxylic acid and 100 µM reduced methanopterin were treated in the same way. The fluorescence emission spectra were recorded at 320, 360 and 395 nm for NXR, methanopterin and pterine-6-carboxylic acid, respectively, on a Cary Eclipse fluorescence spectrophotometer (Agilent Technologies). The excitation spectra were recorded at the respective optimal emission wavelengths of 388, 455 and 436 nm.

**UV–Vis absorption spectroscopy of KsNXR-ABC.** Purified KsNXR-ABC (150 µg) was oxidized under aerobic conditions with 100 µM potassium ferricyanide ($E_0' = +0.43$ V) in 50 mM KP$_i$ buffer, pH 7.25, in 1-ml quartz cuvettes. For reduction of the enzyme, the UV–Vis absorption spectrum (300 to 600 nm) was followed on a Cary 60 spectrophotometer (Agilent Technologies) on addition of the reductants sodium dithionite ($E_0' = -0.66$ V), ascorbic acid ($E_0' = -0.07$ V) or sodium nitrite in excess.

**Spectrophotometric enzyme assays.** All assays were performed at 35 °C using a Cary 60 spectrophotometer (Agilent Technologies). Anaerobic reduction assays

were prepared in an anaerobic glove box (95% Ar/5% H$_2$) and glass cuvettes (1 cm path length) were sealed with Suba-Seal septa (Sigma Aldrich) before transfer to the spectrophotometer. Exetainers were used for aerobic oxidation assays and a Fibre Optic Dip Probe (Agilent Technologies) was inserted directly into the liquid. To assess enzyme kinetics, nonlinear regression analysis was performed using Origin v.9.1 software, applying the Michaelis–Menten equation. The nitrate reduction activity of NXR was determined by following the oxidation of MV ($E_0'$ [MV$_{605\,nm}$] $= -0.45$ V) at 605 nm ($\Delta\varepsilon$ [MV$_{605\,nm}$] $= 13,700$ M⁻¹ cm⁻¹)[42]. The reduced MV monocation radical was prepared by adding dithionite crystals to a 1 mM stock solution of the oxidized compound in 50 mM KP$_i$ buffer, pH 7.2. To avoid excess dithionite, the stock was not completely reduced. Purified KsNXR-ABC (10 µg) was added to 1 ml of reduced MV in KP$_i$ buffer with an absorbance at 605 nm of ~1 and the assay was started with the addition of various concentrations of sodium nitrate (1–100 µM). Alternatively, benzyl viologen ($E_0' = -0.37$ V) and methylene blue ($E_0' = +0.01$ V) were used as artificial electron donors. Sodium selenate, DMSO, sodium thiosulfate, potassium tetrathionate (Fluka) and sodium chlorate were tested as potential substrates (100 µM). To follow nitrite oxidation, purified NXR (30 µg) in 50 mM KP$_i$ buffer, 150 mM NaCl, pH 7.25, was preincubated for 5 min with 800 µM potassium ferricyanide before starting the assay by adding varying amounts of sodium nitrite (1–250 µM). The reduction of ferricyanide was followed at 420 nm ($\Delta\varepsilon$ [ferricyanide$_{420\,nm}$] $= 1$ mM⁻¹ cm⁻¹)[43]. Sodium sulfite and trimethylamine hydrochloride were also tested as potential substrates (100 µM).

**Nitrite conversion and nitrate production rates.** Purified KsNXR-ABC (100 µg) was added to 1 ml of 50 mM KP$_i$ buffer, pH 7.25, containing 800 µM ferricyanide. The reaction was started by the addition of 5 µM sodium nitrite. Every 2 minutes, 50-µl samples were taken and immediately analysed by a Sievers Nitric Oxide (NOA 280i) Analyzer (GE Analytical Instruments), until all nitrite was consumed to a concentration below the detection limit (500 nM).

**Bioinformatics tools.** An in-house protein database was created by translating all open reading frames encoded in the genome (PRJNA16685) of *K. stuttgartiensis*[13]. SignalP v.4.0 (ref. [44]) and TMHMM v.2.0 (ref. [45]) software packages were used to evaluate putative signal peptides and transmembrane helices. For identification of *tat* leader sequences, the tatP software was used[46]. The HHpred homology detection and structure prediction program[47] was used to identify structurally similar proteins. Homologous protein sequences were searched using the Basic Local Alignment Search Tool (BLAST P, http://blast.ncbi.nlm.nih.gov)[48,49] against non-redundant protein sequences from bacteria (taxid: 2) that were retrieved from the National Center for Biotechnology Information (NCBI) databases. Alignments of multiple protein sequences were performed using Clustal Omega (http://www. ebi.ac.uk/Tools/msa/clustalo/)[50] applying default settings. Alignment figures were prepared using ESPript[51].

**KsNXR-ABC quantification, polyacrylamide gel electrophoresis and protein identification.** Protein quantification was performed using the two-dimensional (2D) Quant Kit according to the product manual (GE Healthcare), with bovine serum albumin (BSA) as standard. To assess the purity of NXR, 30 µg of protein was loaded onto a 10% SDS–PAGE gel. The PageRuler Plus Prestained Protein Ladder (Thermo Scientific) was used as molecular weight marker. Subunits were identified by matrix-assisted laser desorption/ionization–time of flight mass spectrometry (MALDI–TOF MS). Gel plugs were picked from visible protein bands and subjected to tryptic digest and MALDI–TOF MS analysis on a Bruker III mass spectrometer (Bruker Daltonik). Peptides were analysed using the Mascot Peptide Fingerprint search software (Matrix Science) against the in-house protein database of *K. stuttgartiensis*. Search settings included carbamidomethylation as a fixed and methionine oxidation as a variable peptide modification. One missed cleavage site and a mass difference of ±0.2 Da were tolerated.

**Crystallization and structure determination of KsNXR-ABC.** KsNXR-ABC was concentrated to an $A_{280}^{1\,cm}$ of 10.5 in 25 mM HEPES/KOH, 25 mM KCl, pH 7.5. Crystals were grown using the hanging-drop vapour diffusion methods by mixing 1 µl of protein stock solution with 1 µl of reservoir solution (50 mM Tris–HCl pH 8.5, 12–14% (w/v) polyethylene glycol 4000, 1.8 mM *n*-decyl-β-D-maltoside) followed by equilibration against 800 µl of reservoir solution. Brown, rod-shaped crystals of ~200 × 50 × 50 µm grew in 2 days and were flash-cooled in liquid nitrogen after soaking in reservoir solution with 20% (v/v) ethylene glycol. Diffraction data were collected at the X10SA beam line of the Swiss Light Source at the Paul Scherrer Institute. Data were integrated with XDS[52], evaluated with CCP4 (ref. [53]) and merged with the STARANISO server[54]. Phasing proved very challenging because the strength of the anomalous signal dropped sharply at low resolution (CC$_{ano}$ < 0.14 at 5 Å despite $R_{pim}$ < 0.05 and $I/\sigma(I)$ > 14 at that resolution). However, SHELXD[55] was able to locate the iron–sulfur clusters as 'superatoms', that is as single scatterers. These appeared to be grouped as eight groups of five iron–sulfur clusters, each of which was reminiscent of the W-shaped constellation of clusters observed in the *E. coli* NarGH complex[29] and the homologous ethylbenzene dehydrogenase from *Aromatoleum aromaticum*[26]. To obtain phases, eight copies of the ethylbenzene dehydrogenase (pdb code 2IVF[26]) were manually

placed into the density, ensuring that their iron–sulfur clusters closely matched the positions of the peaks in an anomalous difference density map calculated using phases from the initial SHELXD solution. Then, rigid body refinement in real space, treating the individual protein subunits as rigid bodies, was used to improve the fit. The resulting iron positions were extracted and given to PHASER for individual heavy atom position refinement and single-wavelength anomalous diffraction (SAD) phasing[56,57]. The resulting phases were then extended to 3.5 Å resolution in very small steps by 1000 cycles of solvent flattening and averaging in PHENIX[58]. This resulted in a map that clearly showed secondary structure elements, and into which an initial model could be built. This model was then refined against the full 3.0-Å resolution data set in PHENIX[58]. Successive cycles of refinement and rebuilding in conventional and averaged maps resulted in a model of excellent quality, with 96.5% of residues in the most-favoured regions of the Ramachandran plot, 3.3% in allowed regions and 0.2% outliers. The final model contains eight KsNXR-ABC heterotrimers. In the subunits of one of the heterotrimers, made up of chains D, E and F in the final model, several regions have poor electron density, and consequently very high B-factors in the model. However, experimental anomalous electron density maps clearly indicate the positions of the cofactors in these subunits, showing that these protein molecules are indeed present in the crystal. This heterotrimer is involved in fewer interactions than the others, so the lower quality of the electron density could be due either to mobility or partial occupancy. Because of the poor electron density, these molecules were excluded from structural analysis. Structural figures were prepared using Pymol (Schrödinger), APBS[59] and ChimeraX[60].

**Sample preparation and data collection for electron cryo-tomography.**
Approximately 100 mg of *K. stuttgartiensis* cells (wet weight) were thawed on ice and resuspended in 20 mM HEPES/NaOH (pH 7.2). Just before freezing, cells were mixed with 10 nm colloidal gold fiducial marker in a 1:1 ratio and 3 µl were applied to freshly glow-discharged R2/2 Cu 300-mesh holey carbon-coated grids (Quantifoil Micro Tools). Grids were plunge-frozen in liquid ethane using a Vitrobot Mark IV plunge-freezer (Thermo Fisher Scientific) at a chamber temperature of 10 °C and a relative humidity of 100%. In total, 18 tomograms with clear visible tubule-like structures inside the anammoxosome were collected on a FEI Titan Krios electron microscope (Thermo Fisher Scientific) operating at 300 kV and equipped with a K2 summit electron detector and quantum energy filter (Gatan). Projections were acquired using SerialEM[61]. Movies were recorded in counting mode at a nominal magnification of ×42,000, which corresponds to a pixel size of 3.585 Å following a dose-symmetric tilt scheme[62] with a 3° increment and a total electron (e−) dose of ~93 e− Å−2.

**Tomogram reconstruction and subtomogram averaging.** Dose-fractionated movies were aligned using MotionCor2 (ref. [63]). Images were combined to generate a raw image stack, which was used as input for IMOD[64]. Single tilt-images were aligned by gold fiducial markers and volumes reconstructed by weighted back-projection. Three-dimensional (3D) contrast transfer function correction was done using NovaCTF[65]. Particle extraction, alignment and subtomogram averaging were performed with Dynamo[66] starting with 7,109 binned particles for first alignment steps. Final refinement of two independent half-maps was done with unbinned data without applying any symmetry. Some 5,126 particles contributed to the final map, which has a resolution of 22 Å at 0.143 FSC (Fourier shell correlation). Volume annotation and segmentation were achieved using tomoseg from the EMAN2 toolbox[67] in combination with ChimeraX[60] for visualization.

**Purification and proteomic analysis of anammoxosomal tubules.** *K. stuttgartiensis* cells (4.85 g wet weight) were resuspended in 20 ml of lysis buffer (150 mM NaCl, 50 mM HEPES/NaOH pH 7.5, 1 mM MgCl₂, 1 mg ml−1 hen egg-white lysozyme, 0.1 mg ml−1 DNAse I, protease inhibitor cocktail (Roche)) and incubated on ice for 30 min after which 100 µM phenylmethylsulfonyl fluoride was added. The sample was then subjected to five cycles of freeze/thaw lysis. The resulting material was centrifuged for 20 min at 1500*g*. The pellet was resuspended in 20 ml of lysis buffer, subjected to another five cycles of freeze/thaw lysis and the lysate was centrifuged as described above. The combined supernatants (40 ml) were centrifuged at 2,000*g* for 20 min at 4 °C, filtered through a 5-µm filter and supplemented with EGTA to a final concentration of 5 mM. After incubation for 12 h on ice, the crude lysate was centrifuged again at 2,000*g* for 20 min at 4 °C and filtered through a 0.8-µm filter. The cleared lysate was concentrated in an Amicon 100-kDa molecular weight cut-off concentrator to ~10 ml and then subjected to size-exclusion chromatography (five separate runs of 2 ml injection volume each) on a Superpose 6 prep grade xk16/70 column using 150 mM NaCl, 50 mM HEPES/NaOH, pH 7.5, at a flow rate of 1 ml min−1 and collecting 1-ml fractions. This resulted in five major peaks that were collected separately. The first peak (40–50 ml elution volume) showed a shoulder and was therefore split into subfractions 1A and 1B. Both of these samples contained the NXR A, B, C and T subunits, as well as the outer membrane porin Kustd1878 as judged by SDS–PAGE and MALDI peptide mass fingerprinting. Previous attempts at purifying tubules that did not use DNAse I in the lysis step also showed the presence of the putative nucleic acid-binding protein Kustc0678 but when a DNAse incubation step was included this protein could no longer be detected.

Negative-stain EM revealed the presence of anammoxosomal tubules as well as membrane fragments, the latter being more prevalent in subfraction 1B. The 1B fraction was therefore subjected to ultracentrifugation on a discontinuous sucrose density gradient (steps of 20, 30, 40, 50, 60 and 70% w/v sucrose in 50 mM HEPES/NaOH pH 7.5, 150 mM NaCl) for 20 h at 4 °C, at 28,000 r.p.m. in an SW28 rotor (Beckman Coulter). This resulted in three main fractions, the top one contained very pure anammoxosomal tubules, as well as fragments thereof, and did not contain detectable amounts of Kustd1878. The fractions from the gradient were collected, dialysed against 50 mM HEPES/NaOH, pH 7.5, 150 mM NaCl and concentrated in Amicon 100 kDa molecular weight cut-off concentrators. The fibre fraction was frozen in small aliquots in liquid nitrogen and stored at −80 °C until further use.

The protein samples were analysed by negative-stain transmission electron microscopy and SDS–PAGE. Tubule proteomics were performed by peptide mass fingerprinting by MALDI–TOF MS on an Axima Performance mass spectrometer (Shimadzu Biotech) using α-cyano-4-hydroxycinnamic acid as the matrix compound. Characteristic tryptic peptides were identified using the MASCOT software (Matrix Science).

**Tubule activity assays.** One-millilitre reaction volumes were prepared in 1-cm path length polystyrene cuvettes, containing 25 µl of density gradient-purified anammoxosomal tubules (43 µg of protein as determined by the Bradford method[68] using BSA albumin as standard) in 50 mM potassium phosphate, pH 7.25, 150 mM NaCl. Assay samples were split into two groups. Dodecyl maltoside (0.5% w/v) was added to one group after which both groups were incubated at 35 °C. Potassium ferricyanide was added to a final concentration of 800 µM and the absorbance at 420 nm was then followed using a Jasco V-650 spectrophotometer (Jasco) until the baseline was stable, after which the reaction was started by adding 2 mM sodium nitrite. Nitrite oxidation rates were calculated from the rate of ferricyanide reduction assuming $\varepsilon_{420} = 1$ mM−1 cm−1 for ferricyanide. Control samples without NXR did not show activity. Rates were determined in triplicate. After the assay, negative-stain electron microscopy was used to assess the number and size of tubules in the assay samples.

**Negative stain transmission electron microscopy.** Protein solutions were diluted to an $A_{280}^{1\,cm}$ of 1–1.5 and then pipetted onto glow-discharged copper grids with carbon-coated formvar (Plano) and stained with aqueous 0.5% (w/v) uranyl acetate solution. Electron micrographs were recorded on an FEI Tecnai G2 T20 twin transmission electron microscope (FEI Nanoport) running at 200 kV accelerating voltage equipped with a FEI Eagle 4k HS, 200-kV charge-coupled device camera.

**Expression and purification of NXR-T (Kustd1705) and Broful_01488.**
The *kustd1705* and *broful_01488* genes were cloned into the custom-designed pUC19Kan2a and pUC19Kan3 vectors[69] to obtain C-terminal and tobacco etch virus (TEV)-protease-cleavable N-terminal His-tagged versions, respectively. The vectors containing the inserts were transformed into *Shewanella oneidensis* MR-1 (triple nuclease mutant ΔendA, ΔexeM, ΔexeS; from Kai Thormann, University of Giessen)[70] by electroporation. For large-scale purification, 10 l of Luria–Bertani medium was supplemented with 50 µg ml−1 kanamycin in 2-l batches in 5-l Erlenmeyer flasks (without baffles) and inoculated with 1% (v/v) of *S. oneidensis* MR-1 overnight culture. Larger cultures were prepared in batches of 10 l or less. The cultures were initially grown at 30 °C at 100 rpm, but after 4.5–5 h, the temperature was lowered to 20 °C at 60 r.p.m. for next 60–70 h. The cells were harvested by centrifugation at 6,000 r.p.m. at 4 °C for 10 min in a Fiberlite F9-6 × 1000 LEX rotor. The *S. oneidensis* cells with expressed proteins were resuspended and homogenized in the wash buffer (50 mM Tris–Cl, pH 8.0, 300 mM NaCl, 10 mM imidazole) in a ratio of 1:3 (w/v), in the presence of one or two 'Complete' protease inhibitor cocktail tablets (Roche Applied Science). The cell suspension was then subjected to sonication at 50% amplitude, 0.5 s 'on' and 'off' times and for total 10 min 'on' time using a Branson W-450 Sonifier (G. Heinemann Ultraschall-und Labortechnik). The cell lysate was then cleared using ultracentrifugation at 45,000 r.p.m. in a Ti-45 rotor for 1 h. The cleared lysate was then loaded onto a Ni-NTA agarose column. The Ni-NTA beads were preincubated with wash buffer. After passing the cleared cell lysate through the column two times, the column was washed twice using ten times the column volume of the wash buffer. The proteins were eluted in elution buffer (50 mM Tris–Cl, pH 8.0, 300 mM NaCl, 300 mM imidazole). After this step, the C-terminally His-tagged proteins were concentrated to a final volume of 1–2 ml using 10-kDa Amicon ultrafilters and subjected to gel-filtration chromatography using a Superdex 75 10/300 column in gel-filtration buffer (50 mM HEPES/NaOH pH 7.5, 150 NaCl). The N-terminally His-tagged proteins, however, were buffer exchanged to TEV digestion buffer (50 mM Tris–Cl pH 8.0, 150 mM NaCl, 2 mM Tris(2-carboxyethyl)phosphine) using 10-kDa Amicon ultrafilters. To 10 ml of buffer-exchanged protein ($A_{280}^{1\,cm} = 2$–3), 500 µl of 1 mg ml−1 N-terminally His-tagged TEV protease was added, and the mixture was incubated overnight in the cold room with stirring. The digested protein was subjected to a second metal-affinity chromatography step using Ni-NTA column material from which the flow-through containing cleaved protein was collected . A cleavage efficiency of 70–80% was observed. The cleaved protein was concentrated using 10-kDa Amicon ultrafilters and subjected to gel-filtration chromatography using Superdex 75 (10/300 GL) column in gel-filtration buffer. Both C-terminally

His-tagged proteins and cleaved proteins were buffer-exchanged to storage buffer (SB; 25 mM HEPES/KOH pH 7.5, 25 mM KCl). The purities of the proteins were analysed by 4–20% SDS–PAGE.

**Analytical ultracentrifugation of NXR-T (Kustd1705) and Broful_01488.**
Sedimentation velocity and sedimentation equilibrium analytical ultracentrifugation were performed as described in ref. [69]. Briefly, the proteins were washed with SB several times on a 10-kDa Amicon ultrafiltration device and loaded into the measurement cell (Beckman Coulter) which was assembled according to the manufacturer's instructions . Samples were sedimented at 25,000 or 30,000 r.p.m. for 24 h at 20 °C in an An-60 Ti rotor in an XL-1 analytical ultracentrifuge (Beckman Coulter). Scans were collected at 415 nm and the data processed with SEDFIT[71].

**UV–Vis spectroscopy of NXR-T (Kustd1705) and Broful_01488.** The as-isolated proteins were diluted in SB to adjust the maximum absorbance at the Soret band to 0.6–0.8. Final protein solutions for spectroscopy were prepared by mixing 135 μl of diluted protein and 15 μl of 100 mM of ligand solution to achieve 10 mM final concentration of the ligands. To measure the as-isolated spectra, 15 μl of SB was added instead of ligand solution. A few grains of sodium dithionite were added to reduce the proteins. The protein solutions were incubated with ligands for at least 15 min at room temperature. Spectra were collected at room temperature using a Jasco V-650 spectrometer (Jasco).

**Crystal structure determination of Broful_01488.** Heterologously produced Broful_01488, after cleavage of the N-terminal His-tag, was crystallized in hanging-drop vapour diffusion setups. The reservoir solution used was 0.02 M lithium chloride, 0.02 M glycine pH 10.0, 33% (w/v) polyethylene glycol 1000 and 0.05 M magnesium chloride. Dark red, rod-shaped crystals were obtained after several days, which were flash-cooled in liquid nitrogen. Diffraction was poor and highly anisotropic; in the best direction, clear Bragg peaks were observed to only ~4 Å resolution. Nevertheless, SAD data were collected at the iron peak wavelength of 1.73 Å and were processed using XDS[52] and STARANISO[54]. We estimate the overall resolution of the data to be 4.0–4.5 Å. Molecular replacement with a single monomer of the KsNXR-T model resulted in the same dimer being found for Broful_01488, but the electron density remained poor. SAD phasing using the haem iron positions from molecular replacement in PHENIX[58], however, did result in an electron density map clearly showing the four-helical fold of the monomers, arranged in the same dimeric configuration as KsNXR-T. A partial model, consisting of four helices, some of the connections between them and the haem group was constructed but could be refined only partially because of the poor quality of the data. The packing of the box-like Broful_01488 dimers in loosely interacting layers of molecules likely explains the low resolution and anisotropy of the diffraction data.

**Determination of the redox potential of NXR-T (Kustd1705).**
Spectropotentiometric measurements were performed at room temperature in a custom-built optically transparent thin-layer electrochemical cell connected to a Keithley model 2450 source measure unit (Tektronix) which was operating as a potentiostat. The electrochemical cell was based on a specially prepared 26 × 76 mm microscopic glass slide. Ultrathin gold electrode connections (<1 μm thickness) were prepared by spray coating with gold paint ('Glanzgold' GG B 15/M; Heraeus) followed by heating to 520 °C. A silver/silver chloride reference electrode patch was prepared using Ag/AgCl ink (ALS). A gold mesh (500 wires per inch, 60% open area, 10 μm thickness; Goodfellow) was used as working electrode. The gold mesh was modified by incubation for >1 h in a solution of 20 mM 4,4′-dithiodipyridine in 160 mM Tris–Cl, pH 8.0 and 20% (v/v) ethanol.

For each measurement, 18 μl of protein solution in 100 mM KCl, 10 mM MOPS/KOH, pH 7.0, was mixed with 2 μl of mediator mix (1 mM each of potassium ferricyanide, p-benzoquinone, 2,5-dimethyl-p-benzoquinone, 1,2-naphtoquinone, phenazine methosulfate, 1,4-napthoquinone, phenazine ethosulfate, 5-hydroxy-1,4-napthoquinone, 2-methyl-1,4-napthoquinone, 2,5-dihydroxy-p-benzoquinone, 2-hydroxy-1,4-napthoquinone, anthraquinone, sodium anthraquinone-2-sulfonate, benzyl viologen and MV). The sample was then placed onto the 4,4′-dithiodipyridine-modified Au-mesh working electrode and covered with a 22 × 22 mm glass cover slip. The cell was then sealed with Parafilm and placed in a Jasco V-760 spectrophotometer. Potentials were set and currents measured using the Keithley source meter using the cyclic voltammetry script provided by the manufacturer. The source meter controlled the spectrophotometer using a custom-built interface. Samples were prepoised at −600 mV for 10 min and then oxidized to +400 mV and reduced back again to −600 mV in 50-mV steps. Spectra were recorded from 350 to 700 nm (90 s per spectrum at 400 nm min⁻¹). Raw spectra were processed using Jasco32 software. Spectra were baseline-corrected setting the absorption at 700 nm to zero. Absorbance values at 415 nm (Soret band) were fitted to a Nernstian function using nonlinear least square minimization in Microsoft Excel to determine the midpoint potential $E_m$:

$$Y = A - A_{ox} = \frac{a}{e^{\left(\frac{nF}{RT}(E - E_m)\right)} + 1} + \text{offset}$$

Here $A$ is the absorbance, $A_{ox}$ the absorbance of the fully oxidized state, $a$ the amplitude, $E$ the potential, $E_m$ the midpoint potential and $n$ the number of electrons ($n = 1$). The Faraday constant $F$ was taken to be 96,485.34 J V⁻¹ mol⁻¹, the temperature $T$ was 293 K and the ideal gas constant $R$ was taken to be 8.3145 J mol⁻¹ K⁻¹. To correct the potential against the SHE, the voltage between the Ag/AgCl patch in a drop of 10 mM MOPS/KOH (pH 7.0), 100 mM KCl and a commercial Ag/AgCl/4 M KCl reference electrode (Pine Research Instrumentation; $E° = +200$ mV versus SHE) was measured.

**In vitro reconstitution of NXR tubules.** Three 50-μl aliquots of purified KsNXR-ABC ($A_{280}^{1\,cm} = 10$) were pooled and diluted to a final volume of ~2 ml in SB. To remove remaining dodecyl maltoside (Supplementary Information), 2 g of Bio-Beads SM (Bio-Rad) were washed two times with 10 ml of SB while stirring for 10 min in a 15-ml falcon tube. The diluted KsNXR-ABC solution was added to the Bio-Beads SM after decanting the excess of SB from the beads. The mixture was then incubated for 2 h at room temperature while stirring. After that, the material was transferred to a 15-ml column while keeping the outlet closed. The flow-through was then collected in a new 15-ml falcon tube. The beads were washed with 5 ml of SB and the wash was collected into the same tube. The resulting protein solution was then concentrated using a 100-kDa cut-off Amicon concentrator to 4 ml.

The concentration of KsNXR-ABC was measured using the Bradford method with BSA as a standard and found to be 3.7 mg ml⁻¹ (or 17.4 μM heterotrimer). Kustd1705 (after cleavage of the N-terminal His-tag) was dispersed at 200 μM (as also determined by the Bradford method) in NXR reconstitution buffer (NRB; 50 mM HEPES/NaOH, pH 7.5, 250 mM NaCl). A 100 mM EGTA stock was prepared by diluting 0.5 M EGTA/NaOH, pH, 8.0 in NRB. For tubule reconstitution, 4 μM of NXR, 20 μM of Kustd1705 (after cleavage of the His-tag) and 10 mM of EGTA in NRB were mixed to a final volume of 10 or 50 μl. The mixture was then incubated in the cold room (6–8 °C) for 24 h before examining tubule formation using negative-stain EM. The reconstitution mixture was then kept in a cold room for 5–6 days and the progress of tubule formation was monitored using negative-stain EM. After 6 days, the tubule-containing mixture was flash-frozen in liquid nitrogen and stored until single particle cryo-EM analysis.

**Cryo-EM of reconstituted KsNXR tubules and helical reconstruction.** Three microlitres of reconstituted tubules were vitrified onto freshly glow-discharged 1.2/1.3 Cu 300-mesh holey carbon-coated grids (Protochips) in liquid ethane using an FEI Vitrobot Mark IV (Thermo Fisher Scientific) at 10 °C and 70% humidity (drain and wait time 0 s, blot force −2 arbitrary units). Data were collected with EPU automatic data acquisition software on an FEI Titan Krios electron microscope (Thermo Fisher Scientific) with a K3 Summit detector (Gatan) operating in counting mode with a calibrated pixel size of 0.837 Å. Total dose was ~50 e⁻ Å⁻² and 50 frames, and defocus values were between −2.0 and −3.0 μm. A total of 1,753 dose-fractionated micrographs were subjected to motion correction and dose-weighting using MotionCor2 (ref. [63]). The micrograph-based CTF was determined by Ctffind with the software package RELION-3.1 (ref. [72]). The resulting images were used for further analysis with RELION-3.1 extended for processing of helical specimens (Extended Data Fig. 6a). Helical segments were picked manually and extracted into boxes of 480 pixels with an overlap of one-third of the box size. This resulted in 9,275 segments. Initial 2D classification produced excellent 2D classes that showed both the unusual architecture of the helical filaments and some secondary structure within the subunits (Extended Data Fig. 6b). For 3D refinement with helical averaging, the model from subtomogram averaging was used as a reference, with varying helical parameters of 100–130 Å (rise) and 100–130° (twist). 3D refinement from 5,379 segments resulted in tubules with higher signal-to-noise ratios with a resolution of 6.2 Å, as assessed by the gold standard FSC (0.143 FSC criterion)[73] and the local resolution was estimated with ResMap (http://resmap.sourceforge.net)[74] (Extended Data Fig. 6c). Density modification with phenix.resolve_cryo_em was carried out using two unmasked half-maps along with the FSC-based resolution and the post processed map[75]. This led to an improvement in the map quality and to a resolution of 5.8 Å (Extended Data Fig. 6d). The structures of NXR-ABC and NXR-T were fitted into the helical reconstruction cryo-EM density map using ChimeraX[60] and adjusted manually (Extended Data Fig. 6e,f).

**Reporting Summary.** Further information on research design is available in the Nature Research Reporting Summary linked to this article.

## Data availability
The NXR-ABC crystal structure and structure factor amplitudes were deposited in the PDB under accession code 7B04. EM maps are available from the EMDB under accession codes EMD-11860 and EMD-11861. Source data are provided with this paper. All other data are available from the authors on request.

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

## Acknowledgements

We thank the staff of the Swiss Light Source (Villigen CH) and the Central Electron Microscopy Facility of the Max Planck Institute of Biophysics (Frankfurt/Main) for their support and facilities, C. Roome, Ö. Yildiz and J. Castillo-Hernández for outstanding computing support and M. Steigleder for expert electronics support. We thank Prof. Dr K. Thormann (University of Giessen) for the kind gift of *S. oneidensis* MR-1. Furthermore, K.P. is very grateful to W. Kühlbrandt for discussions, critical reading of the manuscript and for his continuous support over many years. This work was funded by the Max Planck Society as well as by ERC Consolidator Grant 724362-STePLADDER to T.R.M.B.; ERC 232937, 339880, 854088 and SIAM 024002002 to M.S.M.J., ERC starting grant GreenT 640422 to B.K. and NWO grant VI.Vidi.192.001 to L.v.N.

## Author contributions

T.R.M.B., B.K. and M.S.M.J. conceived the research. N.M. de A. purified KsNXR-ABC, performed biochemical characterization and determined enzyme kinetics. N.M. de A., M.A. and A.D. performed the bioinformatics analysis. T.M.C. and T.R.M.B. determined the crystal structure of KsNXR-ABC. L.D., R.S. and K.P. performed cryo-electron tomography and helical reconstruction EM. M.A. and A.D. crystallized KsNXR-ABC and performed spectropotentiometry. M.A., D.L., A.D. and K.-A.S. performed tubule isolation, characterization and reconstitution. M.M. performed peptide mass fingerprinting. F.L. performed negative-stain electron microscopy. M.A. and K.-A.S. expressed and purified Broful_01488. M.A., E.H. and I.S. crystallized Broful_01488. I.S. collected crystallographic data. G.H.L.N. performed *K. stuttgartiensis* culturing. All authors contributed to data analysis and discussed the results. T.R.M.B. wrote the paper with input from all authors.

## Funding

## Competing interests

The authors declare no competing interests.

## Additional information

**Extended data** is available for this paper at https://doi.org/10.1038/s41564-021-00934-8.

**Correspondence and requests for materials** should be addressed to A.D., B.K., K.P. or T.R.M.B.

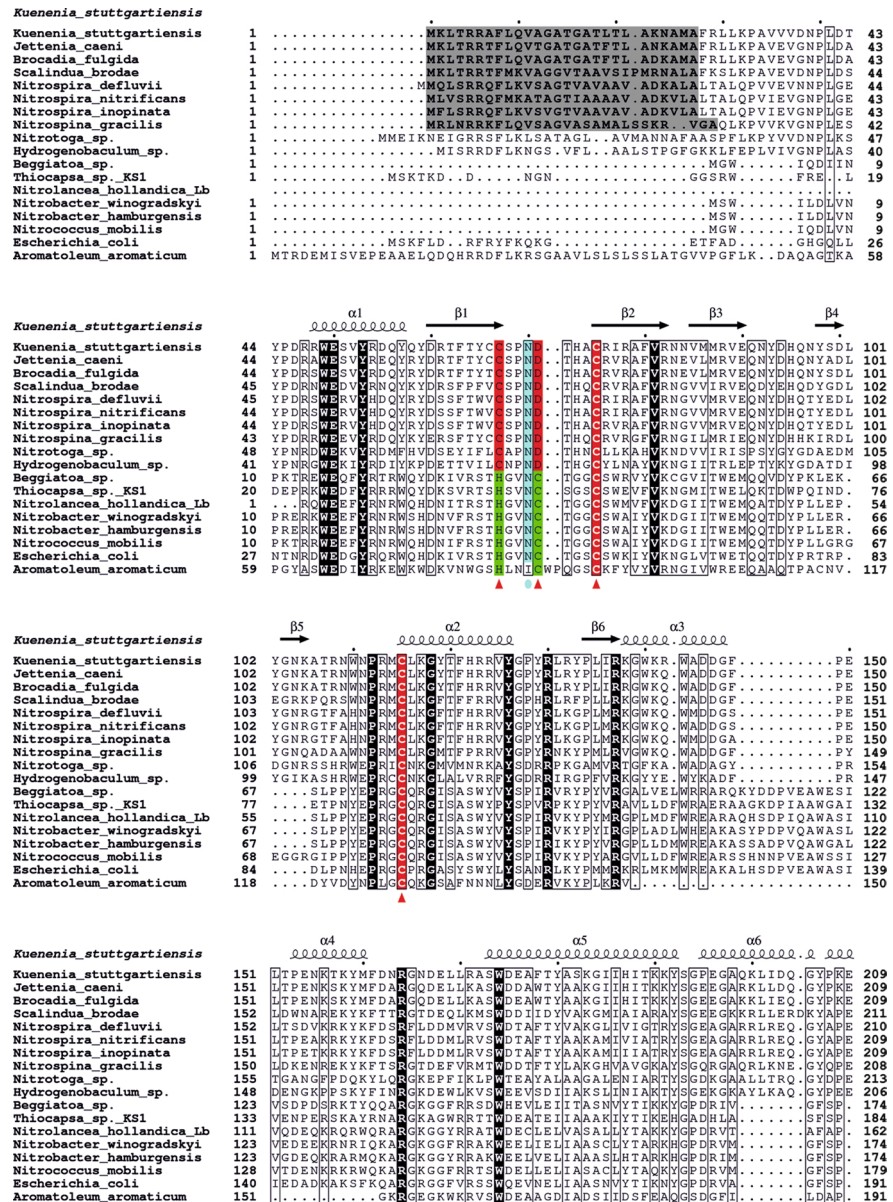

**Extended Data Fig. 1 | Part of the multiple sequence alignment of the catalytic NXR-A subunit.** The sequence of *Kuenenia stuttgartiensis* Kustd1700 (WP_099324707.1) was aligned to its orthologues from *Jettenia caeni* (Planctomycete KSU-1, KSU1_B0257, WP_007220240.1), *Brocadia fulgida* (BROFUL_02528, KKO18624.1) and *Scalindua brodae* (SCABRO_01076, KHE93157.1) using ClustalO[50]. The alignment also contains sequences of the NXR A subunits from the nitrite-oxidizing bacteria (NOB, including comammox species) *Nitrospira defluvii* (WP_013249749.1), *Nitrospira nitrificans* (WP_090900831.1), *Nitrospira inopinata* (WP_062483509.1) and *Nitrospina gracilis* (WP_042250442.1). Moreover, the sequences from *Candidatus* Nitrotoga sp. (MBA0901970.1), the hydrogen-oxidizing bacterium *Hydrogenobaculum* sp. Y04AAS1 (WP_012513384.1) and the nitrate reductase from *Beggiatoa* sp. 4572_84 (OQY53866.1) were aligned. The last group in the alignment comprises the NAR from *Thiocapsa* sp. KS1 (WP_093186287.1) as well as the NXR from *Nitrolancea hollandica* Lb (AFN37206.1) and the nitrate reductases from the *Nitrobacter*-type organisms *Nitrobacter winogradskyi* (WP_011315305.1), *Nitrobacter hamburgensis* (WP_041359063.1), *Nitrococcus mobilis* (WP_004998773.1) which are related to the dissimilatory nitrate reductase from *Escherichia coli* str. K-12 (NP_415742.1) as well as the alpha-subunit of ethylbenzene dehydrogenase (EbdA) from *Aromatoleum aromaticum* (Q5P5I0). *K. stuttgartiensis* NXR A shares about 85% sequence identity with its orthologues from *J. caeni* and *B. fulgida* as well as 66% with *S. brodae*. The sequence identities are around 60% with the NOB and comammox homologues 40% with *Nitrotoga* and *Hydrogenobaculum*, whereas they are only around 20% with the *Nitrobacter*-type enzymes, the nitrate reductases and EbdA. Fully conserved peptide sequences are marked black. The residues binding to the [4Fe-4S] cluster (FS0) are highlighted in red and marked with a red triangle (C67, D71, C75, C115). The predicted tat signal peptides are highlighted in grey, and the proposed substrate-coordinating N70, which is I in ethylbenzene dehydrogenase, is shown in light blue. The figure was prepared using ESPript[51]. The complete alignment is provided as a supplemental file.

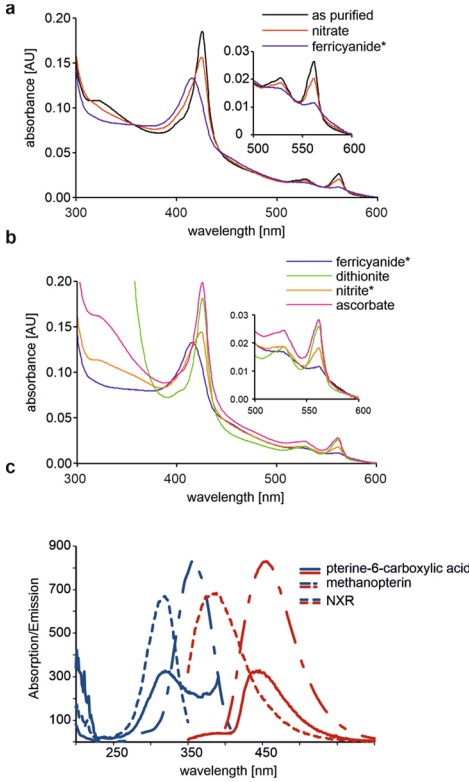

**Extended Data Fig. 2 | Spectroscopy of NXR-ABC.** a, As-purified NXR-ABC showed a reduced heme spectrum. When an excess of sodium nitrate was added, the heme spectrum indicated the heme remained in the reduced state, upon addition of 100 μM potassium ferricyanide, the spectrum indicated the heme was oxidized. b, the heme was oxidized when potassium ferricyanide was added and reduced again by the addition of excess sodium dithionite, sodium nitrite or ascorbic acid. Insets: close-up of alpha- and beta bands. (*: The 100 μM potassium ferricyanide- or nitrite spectrum was subtracted from the heme spectrum.) c Fluorescence excitation (blue) and emission spectra (red) of model compounds and NXR-ABC after permanganate oxidation. NXR, methanopterin and pterin-6-carboxylic acid were oxidized by heat-treatment, followed by the addition of potassium permanganate. The wavelengths recorded were: NXR: 320 nm excitation/388 nm emission; methanopterin: 360 nm excitation/455 nm emission; pterin-6-carboxylic acid: 395 nm excitation/436 nm emission. The NXR sample's excitation and emission spectra are suggestive of a pterin, but do not allow the exact type of pterin to be determined.

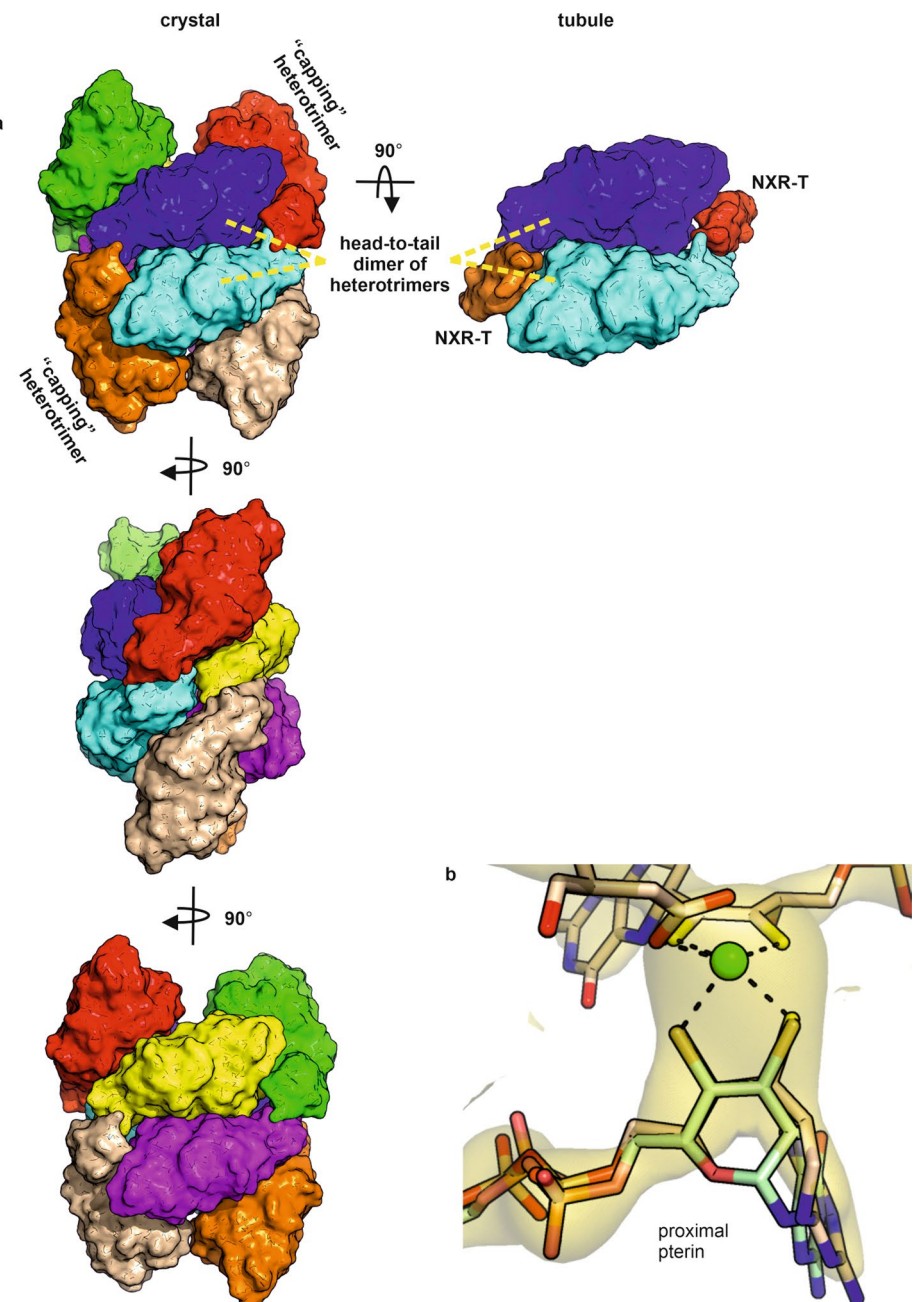

**Extended Data Fig. 3 | Crystallography of NXR-ABC.** a, asymmetric unit of the KsNXR-ABC crystal structure (left, in three orientations) compared with interactions present in the tubules (right). The eight KsNXR-ABC heterotrimers in the crystallographic asymmetric unit are shown in different colors; one strand of the double helix making up the asymmetric unit is shown in orange, blue, light blue and red, the other in beige, yellow, purple and green. Each strand consists of the same head-to-tail dimer of heterotrimers (shown in blue and light blue in the topmost orientation shown) that is also observed in the tubules. The strands of the helices in the crystal structure are capped on either side with a further heterotrimer (orange and red in the topmost orientation) that take the position of KsNXR-T in the tubules (top right). b, the locally averaged simulated annealing 2mFo-DFc omit map is shown (beige) as well as the pterins (beige sticks) and molybdenum atom (green sphere) of the final refined KsNXR-ABC structure. The closed proximal pterin of the EBDH structure[26] is shown in light green. The density shows that like the distal pterin, the proximal pterin is in the pyran-open state.

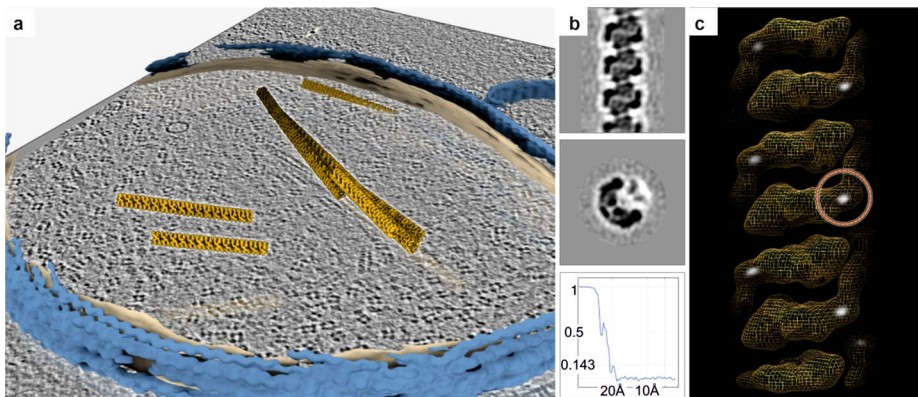

**Extended Data Fig. 4 | Cryo-electron tomography of *Kuenenia stuttgartiensis*.** a, Segmented tomogram of *a K. stuttgartiensis* cell. The S-layer is shown in blue, the anammoxosomal membrane in beige and the tubules in orange. The fourfold-symmetrical particles visible in the slice through the tomogram (grey) are hydrazine dehydrogenase complexes[76]. b, Subtomogram average. The top and middle panels show views perpendicular to and along the tubule axis, respectively, and the bottom panel the FSC curve. c, the highest density (white) in the connecting density corresponds to the heme sites.

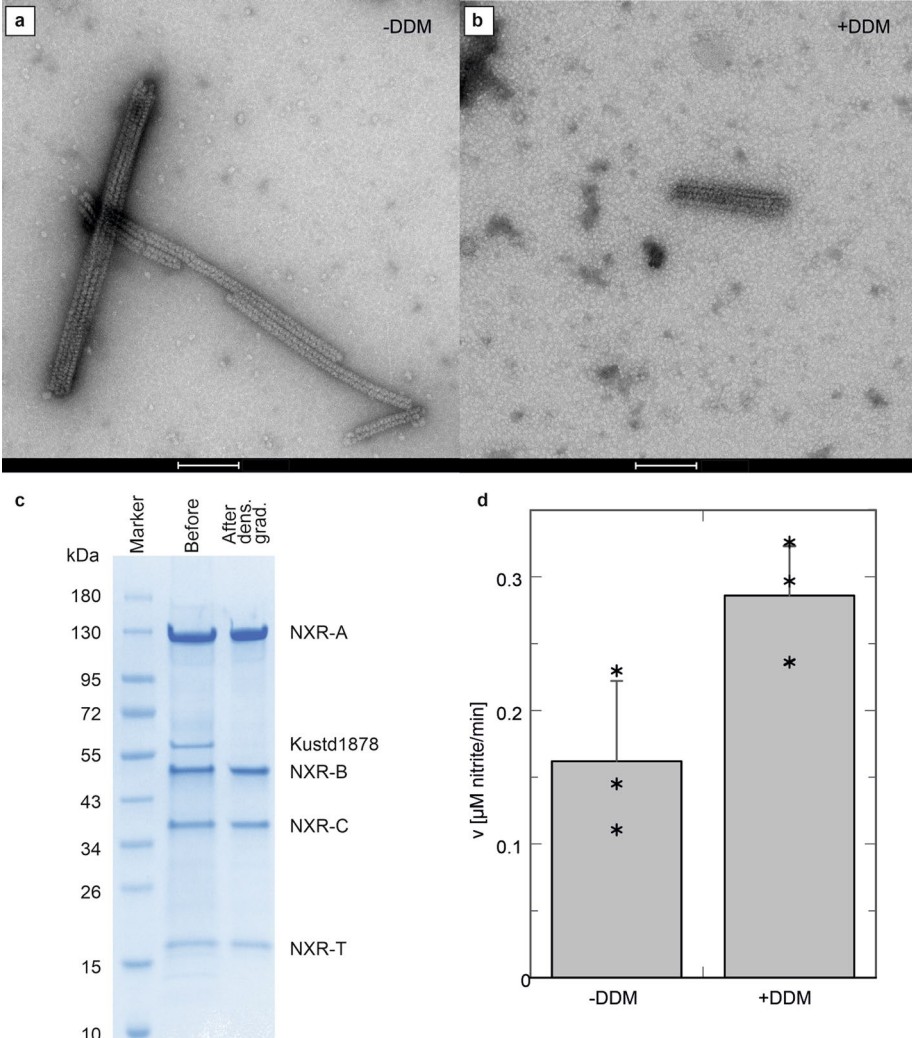

**Extended Data Fig. 5 | Proteomics of anammoxosomal tubules.** a, Negative stain electron micrograph of anammoxosomal tubules purified from *K. stuttgartiensis*. b, as panel a but after incubation with DDM. (scale bars: 100 nm). Shown are representative micrographs from three independent experiments. c, Coomassie-stained SDS-PAGE gel of the tubule preparation. The left lane (M) shows molecular weight markers, with their apparent MW indicated in kDa. The middle and right lanes show the tubule preparation before and after density gradient centrifugation, with the proteins labeled as identified by peptide mass fingerprinting. The outer membrane porin Kustd1878 is no longer present after the centrifugation step, whereas NXR-T is. Tubule purification and analysis was performed twice, showing the presence of NXR-T each time. d, nitrite oxidation assay results on purified anammoxosomal tubules incubated at 35 °C without (-DDM) and with (+DDM) the detergent dodecyl maltoside. Assays were performed in triplicates; the bar graps shows the average, the error bar represents the standard deviation and individual datapoints are shown as asterisks.

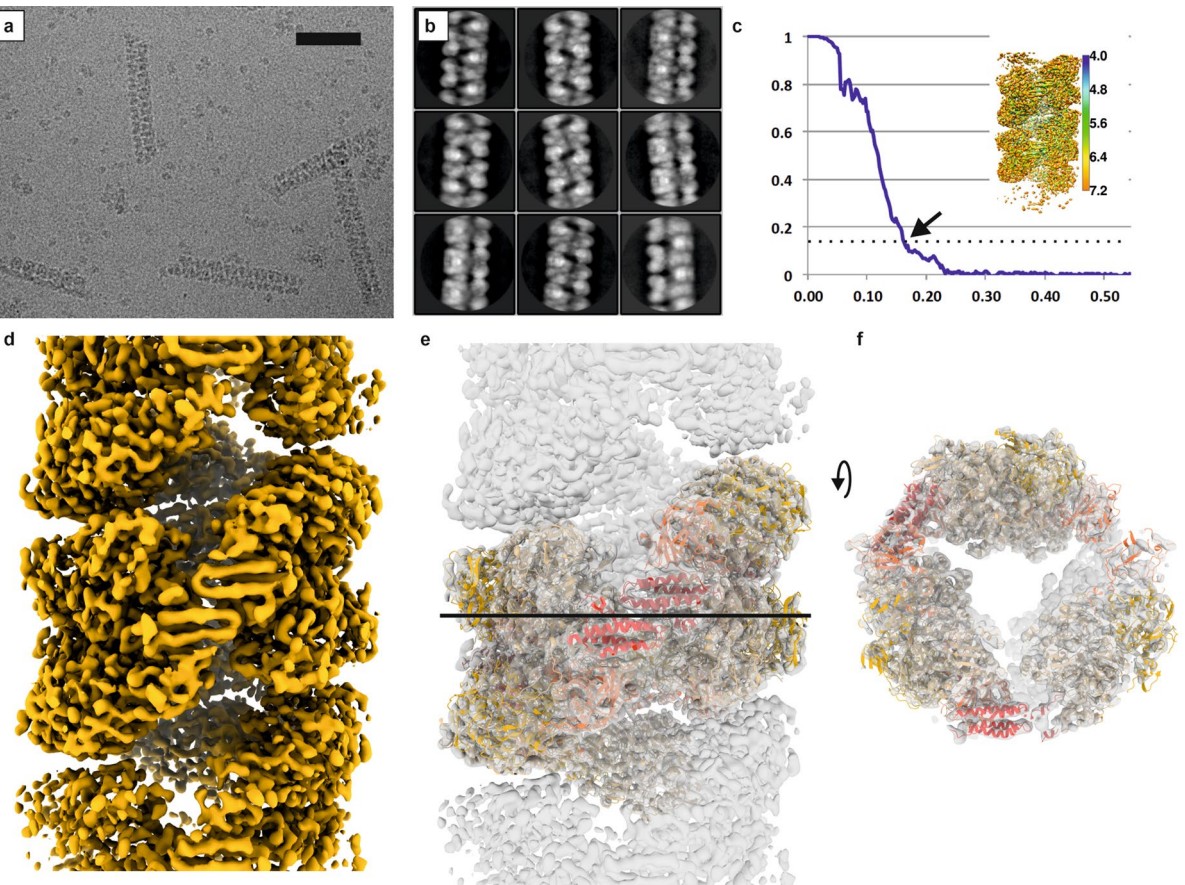

**Extended Data Fig. 6 | Helical reconstruction of the anammoxosomal NXR tubules.** a, Raw micrograph of anammoxosomal tubules, representative of the entire pool of 1,753 micrographs. b, Selection of 2D reference-free class averages of tubules sorted by RELION3.1 in different orientations. c, Gold-standard FSC plots between two separately refined half-maps are shown in blue. The map resolution is indicated by the point where the curve drops below the 0.143 threshold. Inset right: helical reconstruction cryo-EM map of tubules colored by local resolution as analyzed by ResMap[74]. d, Cryo-EM map after density modification with phenix.resolve_cryo_em[75] which improved the map quality to a resolution of 5.8 Å. e, f, Fit of the crystal structures of KsNXR-ABC and KsNXR-T and slice through the cryo-EM map.

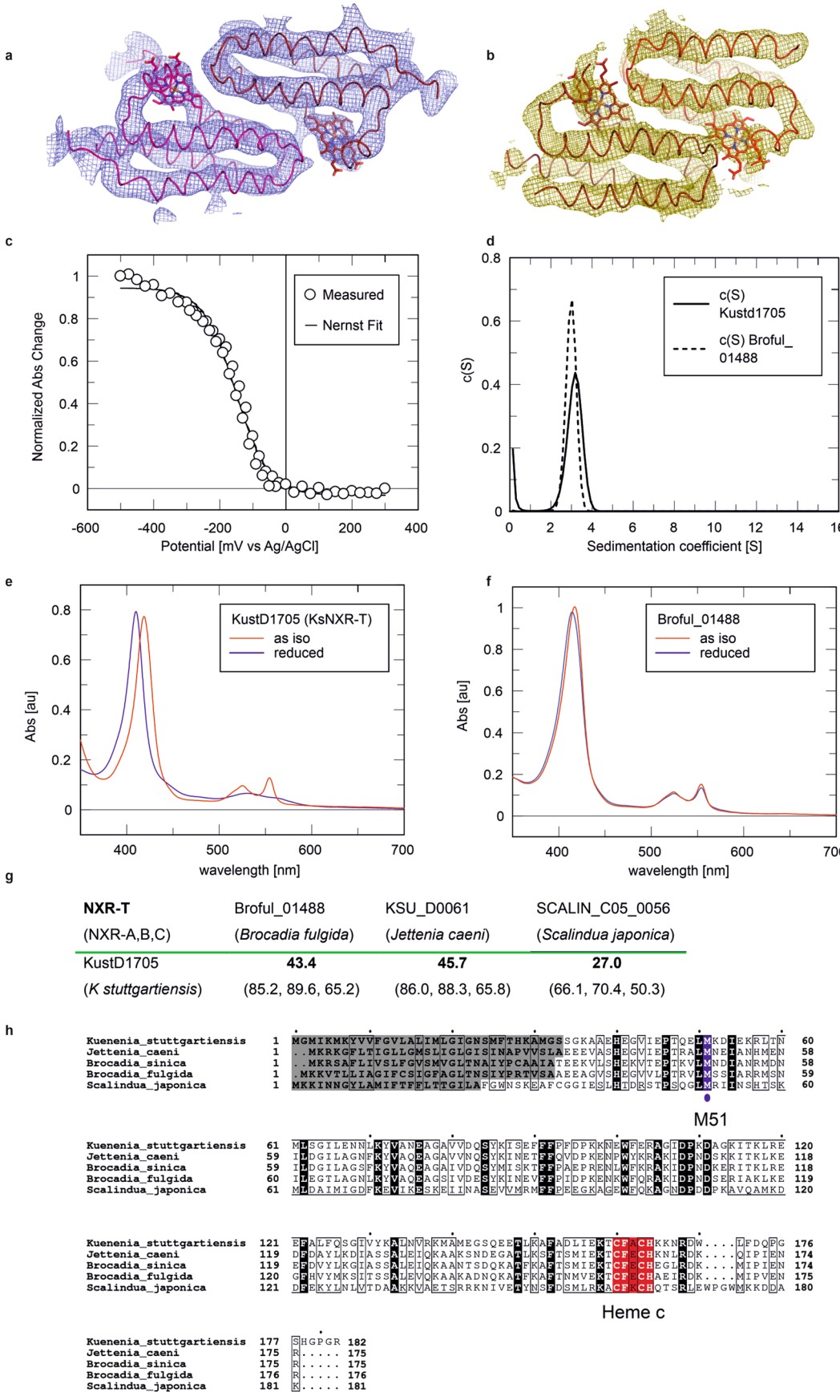

**Extended Data Fig. 7 | See next page for caption.**

**Extended Data Fig. 7 | Analysis of Kustd1705 (KsNXR-T) and Broful_01488.** a, Cryo-EM density from helical reconstruction showing the KsNXR-T (Kustd1705) dimer. A model of Kustd1705, based on the Broful_01488 model (panel B) and refined against the Cryo-EM density, is shown in red. b, 4 Å resolution 2m*Fo*-D*Fc* electron density map contoured at 1 σ of a dimer of Broful_01488, the NXR-T homolog from *Brocadia fulgida*. The proteins show a highly similar, four-helix-bundle fold (red ribbons). Heme groups are shown as sticks. c, Full spectropotentiometric oxidation/reduction trace for Kustd1705 (KsNXR-T), with the fit to the Nernst equation. After correction for the Ag/AgCl reference electrode's potential, a midpoint potential of +120 mV vs SHE is obtained. d, c(S) distributions obtained for Kustd1705 and Broful_01488 by sedimentation velocity analytical ultracentrifugation. e, UV/Vis spectra of as-isolated and reduced Kustd1705. f, UV/Vis spectra of as-isolated and reduced Broful_01488. g, Sequence similarity of NXR-T homologs with Kustd1705. The top numbers (in boldface) show the sequence identity. Below are the sequence identities for the NXR-A,B and C subunits for comparison. h, sequence alignment of NXR-T homologs. The distal methionine is shown in blue, the heme-binding motif in red, and the predicted signal sequence in grey.

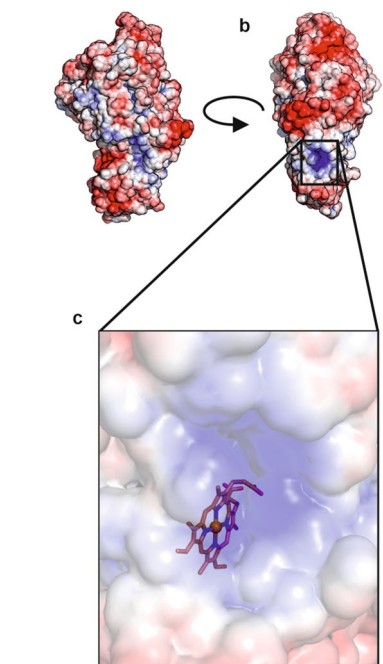

**Extended Data Fig. 8 | Electrostatic surface of the KsNXR-ABC heterotrimer.** The solvent accessible surface is shown, colored from blue (+7 kT/e) to red (−7 kT/e) a,b, overall views of the heterotrimer in two orientations differing by 90°. c, Closeup of the positively charged cleft leading towards the heme *b* binding site.

Boran Kartal
Kristian Parey
Thomas Barends

# Reporting Summary

Nature Research wishes to improve the reproducibility of the work that we publish. This form provides structure for consistency and transparency in reporting. For further information on Nature Research policies, see our Editorial Policies and the Editorial Policy Checklist.

## Statistics

For all statistical analyses, confirm that the following items are present in the figure legend, table legend, main text, or Methods section.

| n/a | Confirmed | |
|---|---|---|
| ☒ | ☐ | The exact sample size (*n*) for each experimental group/condition, given as a discrete number and unit of measurement |
| ☐ | ☒ | A statement on whether measurements were taken from distinct samples or whether the same sample was measured repeatedly |
| ☒ | ☐ | The statistical test(s) used AND whether they are one- or two-sided<br>*Only common tests should be described solely by name; describe more complex techniques in the Methods section.* |
| ☒ | ☐ | A description of all covariates tested |
| ☒ | ☐ | A description of any assumptions or corrections, such as tests of normality and adjustment for multiple comparisons |
| ☒ | ☐ | A full description of the statistical parameters including central tendency (e.g. means) or other basic estimates (e.g. regression coefficient) AND variation (e.g. standard deviation) or associated estimates of uncertainty (e.g. confidence intervals) |
| ☒ | ☐ | For null hypothesis testing, the test statistic (e.g. *F*, *t*, *r*) with confidence intervals, effect sizes, degrees of freedom and *P* value noted<br>*Give P values as exact values whenever suitable.* |
| ☒ | ☐ | For Bayesian analysis, information on the choice of priors and Markov chain Monte Carlo settings |
| ☒ | ☐ | For hierarchical and complex designs, identification of the appropriate level for tests and full reporting of outcomes |
| ☒ | ☐ | Estimates of effect sizes (e.g. Cohen's *d*, Pearson's *r*), indicating how they were calculated |

*Our web collection on statistics for biologists contains articles on many of the points above.*

## Software and code

Policy information about availability of computer code

| Data collection | Crystallographic data collection software: XDS (version 10.1.2014) | CryoEM: MotionCor 1.2.6, IMOD 4.10.9, SerialEM 3.6.22 |
|---|---|
| Data analysis | Crystallographic data analysis software: PHENIX (version 1.14rc1_3177), CCP4 7.1.003, PHASER 2.8.3. EM: NovaCTF 4f134c7, Dynamo 1.1.478, EMAN 2.31, CTFFIND 4.1.13, RELION 3.1. Local density averaging as described in the referenced paper. Figures prepared with Chimera 1.14, Pymol 2.4.1,Origin 9.6.5,ESPript 3.0, GraFit 7.0.0 |

For manuscripts utilizing custom algorithms or software that are central to the research but not yet described in published literature, software must be made available to editors and reviewers. We strongly encourage code deposition in a community repository (e.g. GitHub). See the Nature Research guidelines for submitting code & software for further information.

## Data

Policy information about availability of data

All manuscripts must include a data availability statement. This statement should provide the following information, where applicable:

- Accession codes, unique identifiers, or web links for publicly available datasets
- A list of figures that have associated raw data
- A description of any restrictions on data availability

The NXR-ABC crystal structure and structure factor amplitudes were deposited in the PDB under accession code 7B04. EM maps are available from the EMDB under accession codes EMD-11860 and EMD-11861. These repositories are publicly accessible. Raw data are associated with this paper online for Figure 1b and ED Figure 5c,d.

# Field-specific reporting

Please select the one below that is the best fit for your research. If you are not sure, read the appropriate sections before making your selection.

☒ Life sciences        ☐ Behavioural & social sciences        ☐ Ecological, evolutionary & environmental sciences

For a reference copy of the document with all sections, see nature.com/documents/nr-reporting-summary-flat.pdf

# Life sciences study design

All studies must disclose on these points even when the disclosure is negative.

| | |
|---|---|
| Sample size | Structure determination used all crystallographic and EM data available; other experiments were performed in triplicates which sufficed for statistical significance given the magnitude of the effects observed. |
| Data exclusions | chains D, E and F in the final crystallographic model showed poor electron density and where therefore excluded from structural analysis after structure determination, in compliance with established practice in the field. |
| Replication | We collected several crystallographic datasets which all showed the same data, to comparable resolutions. The kinetic traces shown in Fig.1 are three independent measurements on separate samples. Other experiments were performed in triplicates, all replications were successful |
| Randomization | A 5% 'test set' for cross-validation was defined by random selection of structure factors during the crystallographic analysis as per established practice. For EM, half-datasets were randomly defined for the purpose of Fourier shell correlation calculations, also as per established practice in the field. For other experiments samples were randomly allocated. |
| Blinding | The random selection of test-set reflections for crystallographic cross-validation as well as the allocation to half-datasets for Fourier shell correlation calculation were performed automatically, well before refinement, and structure refinement was done automatically as well. The investigators had no influence over either process, nor where they able to influence the results by 'cherry picking' the data before these procedures started. For other experiments no blinding was necessary as the magnitude of the effect observed precluded inadvertent human influence over the interpretation of the results. |

# Reporting for specific materials, systems and methods

We require information from authors about some types of materials, experimental systems and methods used in many studies. Here, indicate whether each material, system or method listed is relevant to your study. If you are not sure if a list item applies to your research, read the appropriate section before selecting a response.

## Materials & experimental systems

| n/a | Involved in the study |
|---|---|
| ☒ | ☐ Antibodies |
| ☒ | ☐ Eukaryotic cell lines |
| ☒ | ☐ Palaeontology and archaeology |
| ☒ | ☐ Animals and other organisms |
| ☒ | ☐ Human research participants |
| ☒ | ☐ Clinical data |
| ☒ | ☐ Dual use research of concern |

## Methods

| n/a | Involved in the study |
|---|---|
| ☒ | ☐ ChIP-seq |
| ☒ | ☐ Flow cytometry |
| ☒ | ☐ MRI-based neuroimaging |

