## [Peer Review File · Nature Microbiology]

Peer Review Information

Journal: Nature Microbiology

Manuscript Title: Structural and functional characterization of the intracellular filament-forming Nitrite Oxidoreductase multiprotein complex

Corresponding author name(s): Thomas Barends

Reviewer Comments & Decisions:

Decision Letter, initial version:
--

Dear Thomas,

Thank you again for your patience while your manuscript "Architecture of the Intracellular Superstructure-Forming Nitrite Oxidoreductase" was under peer-review at Nature Microbiology. As I explained in my previous email, I had hoped to send you a decision much sooner but we were waiting to hear back from one of the referees, who unfortunately was quite delayed in returning their comments to us. After chasing that referee several times, we still haven't received their report, so at this stage we have decided to nonetheless proceed with the editorial process as we feel that the other 3 referees have the necessary expertise to assess all aspects of the paper.

So your manuscript has now been seen by 3 referees, whose expertise and comments you will find at the end of this email. As you will see from the reports, although they find your work of interest, they have raised a few concerns that will need to be addressed before we can consider publication of the work in Nature Microbiology. In particular, you will see that the two structural referees find the structures well done and informative, and raise only a few points related to some of the methods used that seem straightforward to address. Referee #2, with expertise in nitrification and anammox, is also quite enthusiastic about the findings but raised one main issue related to the need to revise the comparative sequence analysis to include all NXR lineages, and revise the interpretations accordingly. The referee went as far as performing a rough analysis that includes all such lineages and finds that some of the interpretations (they focus for example on residues Asp71, Asn312, Glu527 and Lys917) are potentially impacted by broadening the comparison to other lineages. The rest of the concerns seem clear and straightforward to address. Should further experimental data and text modifications allow you to address these criticisms, we would be happy to look at a revised manuscript.

I'd also like to take a chance to acknowledge your previous email stating that, while the manuscript was in revision, you continued investigating the tubules describe in the manuscript and found out that the protein is enzymatically active in the tubule form; improved the purification protocol; and can now

confirm that the second 'new' protein found in the tubules is indeed a contamination. As mentioned before, we do feel that these points are important to clarify so I'd again recommend adding these to the manuscript during the revision process. We can then ask the referees to specifically comment on these new data when reassessing the new version of the paper.

We are committed to providing a fair and constructive peer-review process so please do not hesitate to contact us if there are specific requests from the reviewers that you believe are technically impossible or unlikely to yield a meaningful outcome.

Please include a data availability statement as a separate section after Methods but before references, under the heading "Data Availability". This section should inform readers about the availability of the data used to support the conclusions of your study. This information includes accession codes to public repositories (data banks for protein, DNA or RNA sequences, microarray, proteomics data etc...), references to source data published alongside the paper, unique identifiers such as URLs to data repository entries, or data set DOIs, and any other statement about data availability. At a minimum, you should include the following statement: "The data that support the findings of this study are available from the corresponding author upon request", mentioning any restrictions on availability. If DOIs are provided, we also strongly encourage including these in the Reference list (authors, title, publisher (repository name), identifier, year). For more guidance on how to write this section please see:

<http://www.nature.com/authors/policies/data/data-availability-statements-data-citations.pdf>

- * Include a "Response to referees" document detailing, point-by-point, how you addressed each referee comment. If no action was taken to address a point, you must provide a compelling argument. This response will be sent back to the referees along with the revised manuscript.
- * If you have not done so already we suggest that you begin to revise your manuscript so that it conforms to our Article format instructions at <http://www.nature.com/nmicrobiol/info/final-submission>. Refer also to any guidelines provided in this letter.
- * Include a revised version of any required reporting checklist. It will be available to referees (and, potentially, statisticians) to aid in their evaluation if the manuscript goes back for peer review. A revised checklist is essential for re-review of the paper.

When submitting the revised version of your manuscript, please pay close attention to our [href="https://www.nature.com/nature-research/editorial-policies/image-integrity">Digital Image Integrity Guidelines. and to the following points below:](https://www.nature.com/nature-research/editorial-policies/image-integrity)

{REDACTED}

Note: This url links to your confidential homepage and associated information about manuscripts you may have submitted or be reviewing for us. If you wish to forward this e-mail to co-authors, please delete this link to your homepage first.

Nature Microbiology is committed to improving transparency in authorship. As part of our efforts in this direction, we are now requesting that all authors identified as 'corresponding author' on published papers create and link their Open Researcher and Contributor Identifier (ORCID) with their account on the Manuscript Tracking System (MTS), prior to acceptance. This applies to primary research papers only. ORCID helps the scientific community achieve unambiguous attribution of all scholarly contributions. You can create and link your ORCID from the home page of the MTS by clicking on 'Modify my Springer Nature account'. For more information please visit www.springernature.com/orcid.

We would hope to receive a revised version of the manuscript within 3 months. If you cannot send it within this time, please let us know. We will be happy to consider your revision, even if a similar study has been accepted for publication at Nature Microbiology or published elsewhere (up to a maximum of 6 months).

In the meantime we hope that you find our referees' comments helpful and please don't hesitate to get in touch if you have any questions.

{REDACTED}

Reviewer Comments:

Reviewer #1 (Remarks to the Author):

Review of Chicano, et al.

This manuscript reports a comprehensive structure-function analysis of nitrite oxidoreductase (NXR)

from the anaerobic ammonium oxidizing bacteria *Kuenenia stuttgartiensis*. The study combines purification from natural sources, mass spectrometric identification of protein components, spectroscopic and spectropotentiometric measurements, x-ray crystallographic structure determination, CryoEM structure determination, and Cryotomographic analysis of in vivo structures of this enzyme complex.

The results are highly novel and significant since this is the first structure of a NXR, which an important reaction to generate bioavailable nitrogen for the Earth's biosphere. The structural and biochemical/biophysical studies reveal several novel findings:

1. The crystal structure of NXRABC show the overall architecture of the complex, unusual features in the active site, and unusual cofactor coordination in the protein's electron transport chain.
2. The previously characterized filamentous anammoxosome identified in cells is composed of NXR.
3. Kustd1705, or NXR-T, was found to be in the filamentous form isolated from cells, and this protein was found to be necessary to assemble filaments of NXRABC in vitro.
4. The structure of a homolog of NXR-T is solved, and shown to fit the region in the helical reconstruction of the filament.
5. The cryoEM work shows how the filament is assembled from the NXRABC protomers involving NXR-T.
6. A likely path for electrons from the active site, to the b-heme of C in a region where binding partners may accept the electrons is proposed.

I have only a few comments and suggestions:

1. AUC data is given in the SI, and a reference to the finding that NXR-T is a dimer in solution is given. My suggestion is to include the sedimentation velocity data (C(s) distributions) in the SI and to give a little more detail in the main text regarding how its dimeric nature was determined.
2. The first sentence of the last paragraph "Such a direct coupling to the nitrite oxidation system explains why in anammox bacteria NXR is located inside the anammoxosome, rather than on a membrane as in NOB, because the NO-producing enzymes in anammox bacteria are soluble, anammoxosomal proteins." Is this sentence proposing a purpose to the filamentation of the enzyme, in order for it to be concentrated relative to freely diffusing enzymes, in a way that being in a membrane might? It's not clear what the meaning of that sentence is, because the next sentence is: "It is not clear at present, however, why *Kuenenia* NXR forms tubules." Is the issue the architecture of the filament (tubular?) or that it forms filaments at all? Clarification of these points would help the reader understand these important take-away messages.
3. Figure 1.B. Regarding the 1:1 stoichiometry, that refers to nitrite and nitrate, correct? Perhaps making that more clear would reduce the possible misinterpretation by the reader.
4. Extended Data Figure 9.A. what is the model shown in the CryoEM map? Is that the same model shown in B? Or is it a refined version of NXR-T based on the structure in B and refinement against the CryoEM data?
5. Figure 3C-D – is there way to indicate the relationship between the views in C and D? I assume it's a rotation about the helical axis by 90 degrees but can't be sure.
6. Extended data figure 3. The two types of dotted lines are a little bit harder than should be necessary to distinguish. Can a different pattern of dashes be used instead?
7. Extended Data, Figure 2. In my printed copy (for what it's worth), the ascorbate line was not visible.
8. Extended Data, Figure 1. I do not see the green markings or triangle. Also, the N70 marked in blue is really hard to see (the amino acid code, that is). Maybe make that text white?
9. Extended Data, Figure 4. The relationship of the colors to 8 copies of ABC heterotrimers is difficult

to see. The coloring is the same in Figure 1C, however, I think in this EDF4 the colors indicate separate heterotrimers, rather than separate chains within a heterotrimer. There are also only 6 separately colored "things" in this figure, are the other two on the opposite side, out of view? The note about how the capping heterotrimers "take the position of KsNXR-T" is also confusing. Are these absent and replaced by KsNXR-T in the filament? It's not mentioned in the text or SI. Also, page 13, in the SI, mention of chains D, E, F having poor electron density. Are these 3 of the 8 heterodimers? Or 3 subunits in one heterodimer? Or 3 subunits from different heterodimers? Clarification on these points would aid the reader.

10. Overall, I think this study provides significant and interesting new information on this important class of enzymes, but the text does feel very brief.

Reviewer #2 (Remarks to the Author):

The manuscript by Chicano et al. reports on the detailed structural analysis of nitrite oxidoreductase (NXR) from an anaerobic ammonium oxidizer, *Kuenenia stuttgartiensis*, and the structure of anammoxosomal tubules formed by NXR and a novel heme c-containing protein termed NxrT.

NXR, which produces most of the nitrate in the biosphere, is a key enzyme of the biogeochemical nitrogen cycle. Different forms of NXR occur in the phylogenetically diverse nitrite-oxidizing bacteria (NOB) and in anammox organisms. Still, there was no structural data for any form of NXR available (only for the related nitrate reductase, NAR, and few other functionally different but structurally related enzymes). That situation was truly embarrassing, as we lacked structural insights into one of the most important enzymes on our planet! The present manuscript closes his knowledge gap for NXR from anammox in a most impressive way. On top of the long-awaited crystal structure of an NxrABC trimer, the authors identified the novel NxrT of anammox that appears to be the 'glue' needed to form the unique NXR tubules of anammox, whose structure was also characterized. The amount of structural data is overwhelming. Like in every great scientific study, some new questions are raised, too. For example: does NxrT serve a function beyond tubule formation? And what is the actual function of the tubular assembly of NXR in anammox? There is absolutely no doubt that this manuscript presents excellent fundamental research and will attract much attention in microbiology, biochemistry, and structural biology.

I read this manuscript with great enthusiasm and found a few minor issues that can easily be addressed. However, I was a bit disappointed that the comparative sequence analysis of NXRs and NARs excludes as many as 4 of the 7 known NXR lineages and at least 1 NAR lineage. As outlined below, this is a major issue and has some potential to change the authors' interpretation of their structural data for NxrA.

Major issue:

There is not just one kind of NXR, but a number of phylogenetic distinct NXR lineages within the type II DMSO reductase superfamily. These are the known NXR lineages (letter designations are arbitrary and not established nomenclature):

a) The membrane-bound, cytoplasmically oriented NXRs of the 'Nitrobacter type' comprising the NXR of *Nitrobacter* and *Nitrococcus*.

b) Related with lineage a, but phylogenetically distinct, is the NXR of Nitrolancea and other Chloroflexi NOB.

c) Another lineage related to a and b is the NXR of the phototrophic NOB, Thiocapsa KS1.

Note: lineages a-c are related with the canonical NAR of *E. coli* and a large number of other membrane-bound NARs from diverse organisms.

d) The membrane-bound or soluble, periplasmically oriented NXRs of the 'Nitrospira type' comprising the NXRs of nitrite-oxidizing and comammox Nitrospira.

e) Related with d, but phylogenetically distinct, is the NXR of anammox.

f) Related with d and e is the periplasmically oriented NXR of Nitrospina.

g) Another distinct lineage of periplasmically oriented enzymes contains the NXR of Nitrotoga, as well as known NARs (e.g., from *Hydrogenobaculum* spp., *Beggiatoa* spp., and other known nitrate reducers). This lineage is more distant from d-f although they share common ancestry. The NARs in this lineage are different enzymes than the canonical NAR of *E. coli*.

Based on their analysis of an anammox NXR in lineage e, the authors discuss structural features of NXR (e.g., in the active site) with respect to their conservation in NXR (versus NAR) and the directionality of the enzymatic reaction. However, this discussion considers only lineages d-f of the periplasmic/anammoxosomal NXRs but does not include the NXR of Nitrotoga (g) and all cytoplasmically oriented NXRs listed above (a-c). The NARs in lineage g are not considered either. This is relevant, because the authors discuss specific residues in NXRs versus their counterparts in NARs at the homologous positions. Such hypotheses on putative key residues of NXR cannot be based on an incomplete analysis.

To illustrate this point, I analyzed an NxrA alignment that includes all lineages. The outcome was:

Asp71 (*K. stuttgartiensis* numbering): This is Asp in NxrA from lineages d-g, but Cys in NxrA from lineages a-c. Thus, this residue does not have to be Asp in an enzyme that functions mainly in the oxidative direction. Moreover, the residue is Asp in NarG from lineage g (*Hydrogenobaculum*, *Beggiatoa*) and Cys in NarG of *E. coli*. The fact that phylogenetically related NXRs and NARs can share the same residue at this position raises doubts on the proposed contribution to the catalytic bias (p.3, l. 33-37). Actually, this observation points more at a phylogenetic signal rather than a functional signal here; there may of course be a functional context as well, but this remains completely unresolved.

Asn312: This is Asn in NxrA from lineages d-f, but it is Thr in NxrA from lineages a-c and g. It is Thr in the NARs, too. Thus, the statement that Asn would be conserved in NXRs (Supplement line 203) is incorrect. Again, this seems to be a phylogenetic signal as lineages d-f are closely related.

Glu527: In contrast to the statement in the Supplement (line 185-186), this residue is not conserved in NXRs. It is Glu in lineages d-f, but Ala in lineage a, b, and g; in lineage c it is Ser. In NARs it is not restricted to Ala or Gly, as it differs in NARs of lineage g.

Lys917 (Supplement lines 203-206): This is Lys in NXR from lineages d-g but His in NXR from lineages a-c. In addition, it is also Lys in NARs of lineage g, suggesting that there is limited influence of this residue on the directionality of the reaction. Again, a phylogenetic signal appears to be likely here, because lineages d-g represent periplasmic NXR and NARs that are more closely related with each other than with the cytoplasmic lineages a-c.

For sequence alignment I used mafft with the strategy L-INS-i, and some details might differ if another alignment method is used. In any case, the authors should reevaluate the conservation patterns and possible roles of the listed residues based on a complete dataset, which should contain NXR and NARs from all lineages. If other putative key residues emerge from such an extended analysis and the NXR structure from anammox, this would certainly be a valuable new result. In any case, the discussion of the various residues should be adapted accordingly, and the alignment in ED Fig. 1 must be extended to include representative NXR and NARs from all lineages (otherwise, this analysis is not sufficiently informative). Moreover, most of the discussed positions (Asn312, Glu527, Lys917) are not included in ED Fig. 1. The figure should be extended to cover those positions, too.

In the introduction part, the authors should mention the phylogenetic diversity of NXR and indicate that the structurally analyzed anammox NXR represents just one lineage of this diversity. This important information will not affect the high scientific value of the results, but it will help readers put the data into perspective and follow an extended analysis and discussion of putative key residues. Useful references for the NXR phylogeny would be: 10.1073/pnas.1003860107 (lineages a,d,e); ref. 20 (lineage f); 10.1038/ismej.2012.70 (lineage b); 10.1038/ismej.2016.56 (lineage c); 10.1128/mBio.01186-18 (lineage g).

Minor issues:

1. p. 1, l. 27: To my knowledge, NXR is not a multienzyme complex. Please rephrase.
2. p. 1, l. 30: Replace 'nitrogen' with 'dinitrogen gas'.
3. p. 2, l. 12: Recent genomic and transcriptomic data suggested that some NXRs from NOB (*Nitrospira moscoviensis*, *Nitrotoga fabula*) may not be membrane-bound but could be soluble in the periplasmic space. References: 10.3389/fmicb.2019.01325 and 10.1128/mBio.01186-18.
4. p. 2, l. 21-22: It would be appropriate to cite the primary study that demonstrated the bidirectional activity of NXR in *Nitrospira* cells: 10.1073/pnas.1506533112.
5. p.3, l. 2: See item #3 for suitable references (instead of ref. 1, which does not address solubility of NXR).
6. p. 3, l. 32-37: See above; in addition, please mention that the presence of Asp at this position in the NXR of anammox and *Nitrospira* was already noticed elsewhere (10.1073/pnas.1003860107).
7. p. 4, l. 6: Figure 3 E+D
8. p. 4, l. 33-38: The NO-generating machinery/enzymes of anammox are repeatedly mentioned here, but more information is missing. Please provide at least the name(s) of those enzyme(s) to assist less

well-informed readers at this point.

9. ED Figure 1: See above; in addition, please add residue numbers also on the right side of each alignment row. The caption mentions Asp275 highlighted in green, but this position is not part of the alignment shown.

10. ED Figure 3 (caption): The sentence "Detection of pterin in KsNXR-ABC" is redundant.

11. Supplement line 164: Should read "KsNXR-ABC".

Reviewer #3 (Remarks to the Author):

This paper reports the structure of a nitrite oxidoreductase multienzyme complex, using a hybrid approach that includes X-ray crystallography, cryo-electron tomography with subvolume averaging, and single particle cryo-EM helical reconstruction. The crystal structure reveals an electron transport path within the NRX-ABC trimer. Cryo-electron tomography of intact bacteria revealed the ultrastructure of the helical multienzyme complex, the "anammoxosome" The work reveals the identity of a third protein, NRX-T, which is essential for formation of the anammoxosome, which allowed reconstitution of the filaments for single particle cryo-EM work. While the resolution of the cryo-EM reconstruction is modest, combining this structure with the crystal structures of NRX-ABC and an NRX-T homolog provided insight into the assembly mechanisms.

Overall, the paper is well written and clear, and reveals the structure of a fascinating multienzyme assembly. It also raises interesting questions about the mechanisms of nitrate generation. The structural work is well done, and the conclusions well supported by the structures.

I have only a few minor comments with respect to the helical reconstruction:

- 1) was the two-fold dyad symmetry, perpendicular to the helical axis, enforced during structure refinement? This isn't mentioned in the methods. If so this should be mentioned, if not it really ought to be as this would very likely improve the resolution of the structure.
- 2) the overlap between adjacent particles extracted from helices should be stated
- 3) what were the final refined helical symmetry parameters (rise/twist per subunit) for this reconstruction? Did these differ from the initial estimate from the subvolume average structure?
- 4) are there direct contacts between NXR-ABC trimers along the length of the helical axis? If so, these should be described, as well as whether they resemble any of the crystal packing interfaces observed in the crystal asymmetric unit.

Author Rebuttal to Initial comments

Reviewer #1 (Remarks to the Author):

Review of Chicano, et al.

This manuscript reports a comprehensive structure-function analysis of nitrite oxidoreductase

(NXR) from the anaerobic ammonium oxidizing bacteria *Kuenenia stuttgartiensis*. The study combines purification from natural sources, mass spectrometric identification of protein components, spectroscopic and spectropotentiometric measurements, x-ray crystallographic structure determination, CryoEM structure determination, and Cryotomographic analysis of in vivo structures of this enzyme complex.

The results are highly novel and significant since this is the first structure of a NXR, which an important reaction to generate bioavailable nitrogen for the Earth's biosphere. The structural and biochemical/biophysical studies reveal several novel findings:

1. The crystal structure of NXRABC show the overall architecture of the complex, unusual features in the active site, and unusual cofactor coordination in the protein's electron transport chain.
2. The previously characterized filamentous anammoxosome identified in cells is composed of NXR.
3. Kustd1705, or NXR-T, was found to be in the filamentous form isolated from cells, and this protein was found to be necessary to assemble filaments of NXRABC in vitro.
4. The structure of a homolog of NXR-T is solved, and shown to fit the region in the helical reconstruction of the filament.
5. The cryoEM work shows how the filament is assembled from the NXRABC protomers involving NXR-T.
6. A likely path for electrons from the active site, to the b-heme of C in a region where binding partners may accept the electrons is proposed.

I have only a few comments and suggestions:

1. AUC data is given in the SI, and a reference to the finding that NXR-T is a dimer in solution is given. My suggestion is to include the sedimentation velocity data ($C(s)$ distributions) in the SI and to give a little more detail in the main text regarding how its dimeric nature was determined.

We thank the referee for this suggestion and have added the $c(S)$ distributions to Extended Data Figure 10, and added a detailed explanation of how these results suggest a dimeric structure in solution in the Supplementary Information. We have also calculated predicted sedimentation coefficients using the program HYDROPRO from the dimeric structures we determined of Kustd1705 and its homolog and the values are consistent with the observed values, further supporting the notion that the proteins are dimeric in solution, too. As these points are rather technical, however, we feel that the supplement would be a better place for these discussions than the main text.

2. The first sentence of the last paragraph "Such a direct coupling to the nitrite oxidation system explains why in anammox bacteria NXR is located inside the anammoxosome, rather than on a membrane as in NOB, because the NO-producing enzymes in anammox bacteria are soluble, anammoxosomal proteins." Is this sentence proposing a purpose to the filamentation of the

enzyme, in order for it to be concentrated relative to freely diffusing enzymes, in a way that being in a membrane might? It's not clear what the meaning of that sentence is, because the next sentence is: "It is not clear at present, however, why Kuenenia NXR forms tubules." Is the issue the architecture of the filament (tubular?) or that it forms filaments at all? Clarification of these points would help the reader understand these important take-away messages.

We apologize; indeed, the structure of our manuscript was rather vague here and we have rewritten this part. Among other things we have separated this into two paragraphs. The first is exclusively concerned with the localization of the enzyme inside the anammoxosome, not with the oligomerization. The next paragraph is then concerned with the tubules and more specifically with possible roles of NXR-T beyond tubule formation. This suggests directions for further research into the role of the tubules and the novel NXR-T subunit. We also explicitly separate the two issues (localization and oligomerization) at the beginning of the last paragraph. We hope that by separating these issues (localization and tubule formation) the text has become sufficiently clear.

3. Figure 1.B. Regarding the 1:1 stoichiometry, that refers to nitrite and nitrate, correct? Perhaps making that more clear would reduce the possible misinterpretation by the reader.

We thank the referee and have amended the figure legend to explicitly state that this concerns nitrite-to-nitrate stoichiometry.

4. Extended Data Figure 9.A. what is the model shown in the CryoEM map? Is that the same model shown in B? Or is it a refined version of NXR-T based on the structure in B and refinement against the CryoEM data?

We thank the referee for picking this up: indeed, this is a model of Kustd1705 based on the Broful_01488 structure (in panel B) which was then refined against the EM data. We have added this information to the figure legend.

5. Figure 3C-D – is there way to indicate the relationship between the views in C and D? I assume it's a rotation about the helical axis by 90 degrees but can't be sure.

The relationship is a 70-degree rotation around the helical axis; we added this information to the figure.

6. Extended data figure 3. The two types of dotted lines are a little bit harder than should be necessary to distinguish. Can a different pattern of dashes be used instead?

We thank the referee for pointing this out and have changed one of the sets of lines to dot-dash lines to clearly set them apart from the solid and dashed sets.

7. Extended Data, Figure 2. In my printed copy (for what it's worth), the ascorbate line was not visible.

The referee is right, this figure was needlessly difficult to look at as it was only in shades of gray. We've made a colored version of this figure.

8. Extended Data, Figure 1. I do not see the green markings or triangle. Also, the N70 marked in blue is really hard to see (the amino acid code, that is). Maybe make that text white?

In response to the issues raised by referee 2 we have prepared a new alignment figure; we have taken care that the same issues don't affect this new figure.

9. Extended Data, Figure 4. The relationship of the colors to 8 copies of ABC heterotrimers is difficult to see. The coloring is the same in Figure 1C, however, I think in this EDF4 the colors indicate separate heterotrimers, rather than separate chains within a heterotrimer. There are also only 6 separately colored "things" in this figure, are the other two on the opposite side, out of view? The note about how the capping heterotrimers "take the position of KsNXR-T" is also confusing. Are these absent and replaced by KsNXr-T in the filament? It's not mentioned in the text or SI.

We have prepared a completely new version of this figure in which each heterotrimer has its own color, and which shows the asymmetric unit of the crystal in three orientations, so as to better illustrate what is going on. We feel this figure is now much clearer.

Also, page 13, in the SI, mention of chains D, E, F having poor electron density. Are these 3 of the 8 heterodimers? Or 3 subunits in one heterodimer? Or 3 subunits from different heterodimers? Clarification on these points would aid the reader.

The referee touches on an important point here: these chains are indeed the subunits of a single heterotrimer, which is located on the edge of the asymmetric unit and is involved in relatively few interactions. This suggest that their poor electron density could be due to either partial occupancy or mobility (or both); we now indicate this in the supplement.

10. Overall, I think this study provides significant and interesting new information on this important class of enzymes, but the text does feel very brief.

We thank the referee very much; we have also slightly expanded the text at the end by speculating on a possible role for the NXR-T subunit.

Reviewer #2 (Remarks to the Author):

The manuscript by Chicano et al. reports on the detailed structural analysis of nitrite oxidoreductase (NXR) from an anaerobic ammonium oxidizer, *Kuenenia stuttgartiensis*, and the structure of anammoxosomal tubules formed by NXR and a novel heme c-containing protein termed NxrT.

NXR, which produces most of the nitrate in the biosphere, is a key enzyme of the biogeochemical nitrogen cycle. Different forms of NXR occur in the phylogenetically diverse nitrite-oxidizing bacteria (NOB) and in anammox organisms. Still, there was no structural data for any form of NXR available (only for the related nitrate reductase, NAR, and few other functionally different but structurally related enzymes). That situation was truly embarrassing, as we lacked structural insights into one of the most important enzymes on our planet! The present manuscript closes his knowledge gap for NXR from anammox in a most impressive way. On top of the long-awaited crystal structure of an NxrABC trimer, the authors identified the novel NxrT of anammox that appears to be the 'glue' needed to form the unique NXR tubules of anammox, whose structure was also characterized. The amount of structural data is overwhelming. Like in every great scientific study, some new questions are raised, too. For example: does NxrT serve a function beyond tubule formation? And what is the actual function of the tubular assembly of NXR in anammox? There is absolutely no doubt that this manuscript presents excellent fundamental research and will attract much attention in microbiology, biochemistry, and structural biology.

I read this manuscript with great enthusiasm and found a few minor issues that can easily be addressed. However, I was a bit disappointed that the comparative sequence analysis of NXRs and NARs excludes as many as 4 of the 7 known NXR lineages and at least 1 NAR lineage. As outlined below, this is a major issue and has some potential to change the authors' interpretation of their structural data for NxrA.

Major issue:

There is not just one kind of NXR, but a number of phylogenetic distinct NXR lineages within the type II DMSO reductase superfamily. These are the known NXR lineages (letter designations are arbitrary and not established nomenclature):

a) The membrane-bound, cytoplasmically oriented NXRs of the 'Nitrobacter type' comprising the NXR of *Nitrobacter* and *Nitrococcus*.

b) Related with lineage a, but phylogenetically distinct, is the NXR of *Nitrolancea* and other

Chloroflexi NOB.

c) Another lineage related to a and b is the NXR of the phototrophic NOB, Thiocapsa KS1.

Note: lineages a-c are related with the canonical NAR of *E. coli* and a large number of other membrane-bound NARs from diverse organisms.

d) The membrane-bound or soluble, periplasmically oriented NXRs of the 'Nitrospira type' comprising the NXRs of nitrite-oxidizing and comammox Nitrospira.

e) Related with d, but phylogenetically distinct, is the NXR of anammox.

f) Related with d and e is the periplasmically oriented NXR of Nitrospina.

g) Another distinct lineage of periplasmically oriented enzymes contains the NXR of Nitrotoga, as well as known NARs (e.g., from *Hydrogenobaculum* spp., *Beggiatoa* spp., and other known nitrate reducers). This lineage is more distant from d-f although they share common ancestry. The NARs in this lineage are different enzymes than the canonical NAR of *E. coli*.

Based on their analysis of an anammox NXR in lineage e, the authors discuss structural features of NXR (e.g., in the active site) with respect to their conservation in NXR (versus NAR) and the directionality of the enzymatic reaction. However, this discussion considers only lineages d-f of the periplasmic/anammoxosomal NXRs but does not include the NXR of Nitrotoga (g) and all cytoplasmically oriented NXRs listed above (a-c). The NARs in lineage g are not considered either. This is relevant, because the authors discuss specific residues in NXRs versus their counterparts in NARs at the homologous positions. Such hypotheses on putative key residues of NXR cannot be based on an incomplete analysis.

We are extremely grateful to the referee for her/his help with this issue! We have added a paragraph describing the current state of knowledge on the phylogeny of NXRs and their relationship with the various types of NARs. Moreover, we have prepared a new version of the alignment figure to include all lineages of NXR and NAR.

To illustrate this point, I analyzed an NxrA alignment that includes all lineages. The outcome was:

Asp71 (*K. stuttgartiensis* numbering): This is Asp in NxrA from lineages d-g, but Cys in NxrA from lineages a-c. Thus, this residue does not have to be Asp in an enzyme that functions mainly in the oxidative direction. Moreover, the residue is Asp in NarG from lineage g (*Hydrogenobaculum*, *Beggiatoa*) and Cys in NarG of *E. coli*. The fact that phylogenetically

related NXR and NARs can share the same residue at this position raises doubts on the proposed contribution to the catalytic bias (p.3, l. 33-37). Actually, this observation points more at a phylogenetic signal rather than a functional signal here; there may of course be a functional context as well, but this remains completely unresolved.

The referee is right; we have changed the text accordingly

Asn312: This is Asn in NxrA from lineages d-f, but it is Thr in NxrA from lineages a-c and g. It is Thr in the NARs, too. Thus, the statement that Asn would be conserved in NXRs (Supplement line 203) is incorrect. Again, this seems to be a phylogenetic signal as lineages d-f are closely related.

Glu527: In contrast to the statement in the Supplement (line 185-186), this residue is not conserved in NXRs. It is Glu in lineages d-f, but Ala in lineage a, b, and g; in lineage c it is Ser. In NARs it is not restricted to Ala or Gly, as it differs in NARs of lineage g.

In addition, it is also Lys in NARs of lineage g, suggesting that there is limited influence of this residue on the directionality of the reaction. Again, a phylogenetic signal appears to be likely here, because lineages d-g represent periplasmic NXRs and NARs that are more closely related with each other than with the cytoplasmic lineages a-c.

We have removed the incorrect statements concerning conservation. However, some parts of these statements are structural descriptions, having no bearing on the discussion of the directionality at all, so we have not removed these.

For sequence alignment I used mafft with the strategy L-INS-i, and some details might differ if another alignment method is used. In any case, the authors should reevaluate the conservation patterns and possible roles of the listed residues based on a complete dataset, which should contain NXRs and NARs from all lineages. If other putative key residues emerge from such an extended analysis and the NXR structure from anammox, this would certainly be a valuable new result. In any case, the discussion of the various residues should be adapted accordingly, and the alignment in ED Fig. 1 must be extended to include representative NXRs and NARs from all lineages (otherwise, this analysis is not sufficiently informative). Moreover, most of the discussed positions (Asn312, Glu527, Lys917) are not included in ED Fig. 1. The figure should be extended to cover those positions, too.

We have changed the text as described above; again, we are enormously grateful to the referee for having done this!

In the introduction part, the authors should mention the phylogenetic diversity of NXR and indicate that the structurally analyzed anammox NXR represents just one lineage of this diversity. This important information will not affect the high scientific value of the results, but it will help readers put the data into perspective and follow an extended analysis and discussion of putative key residues. Useful references for the NXR phylogeny would be: 10.1073/pnas.1003860107 (lineages a,d,e); ref. 20 (lineage f); 10.1038/ismej.2012.70 (lineage b); 10.1038/ismej.2016.56 (lineage c); 10.1128/mBio.01186-18 (lineage g).

As stated above, we have added a paragraph discussing this.

Minor issues:

1. p. 1, l. 27: To my knowledge, NXR is not a multienzyme complex. Please rephrase.

The referee is right; we have changed this to multiprotein complex.

2. p. 1, l. 30: Replace 'nitrogen' with 'dinitrogen gas'.

This is indeed much better; we've made the change as suggested.

3. p. 2, l. 12: Recent genomic and transcriptomic data suggested that some NXRs from NOB (Nitrospira moscoviensis, Nitrotoga fabula) may not be membrane-bound but could be soluble in the periplasmic space. References: 10.3389/fmicb.2019.01325 and 10.1128/mBio.01186-18.

We thank the referee and have added this information and the references to the main text.

4. p. 2, l. 21-22: It would be appropriate to cite the primary study that demonstrated the bidirectional activity of NXR in Nitrospira cells: 10.1073/pnas.1506533112.

We agree completely with the referee and have added this reference to the main text.

5. p.3, l. 2: See item #3 for suitable references (instead of ref. 1, which does not address solubility of NXR).

We have exchanged reference 1 for correct references as pointed out.

6. p. 3, l. 32-37: See above; in addition, please mention that the presence of Asp at this position in the NXR of anammox and Nitrospira was already noticed elsewhere (10.1073/pnas.1003860107).

We have added this information to the MS, thank you.

7. p. 4, l. 6: Figure 3 E+D

We thank the referee and have correct the figure reference.

8. p. 4, l. 33-38: The NO-generating machinery/enzymes of anammox are repeatedly mentioned here, but more information is missing. Please provide at least the name(s) of those enzyme(s) to assist less well-informed readers at this point.

The text now explicitly mentions nitrite reductase as the NO-generating enzyme, as well as its function.

9. ED Figure 1: See above; in addition, please add residue numbers also on the right side of each alignment row. The caption mentions Asp275 highlighted in green, but this position is not part of the alignment shown.

We have prepared a completely new version of this figure as set out above, and added residue numbers on the right side as requested. For space reasons ED Figure 1 contains only the first part of this alignment; the rest of the alignment is now provided as a supplementary file.

10. ED Figure 3 (caption): The sentence "Detection of pterin in KsNXR-ABC" is redundant.

We thank the referee for pointing this out and have corrected the error.

11. Supplement line 164: Should read "KsNXR-ABC".

We thank the referee for picking this up and have corrected this error, too.

Reviewer #3 (Remarks to the Author):

This paper reports the structure of a nitrite oxidoreductase multienzyme complex, using a hybrid approach that includes X-ray crystallography, cryo-electron tomography with subvolume averaging, and single particle cryo-EM helical reconstruction. The crystal structure reveals an electron transport path within the NRX-ABC trimer. Cryo-electron tomography of intact bacteria revealed the ultrastructure of the helical multienzyme complex, the "anammoxosome" The work reveals the identity of a third protein, NRX-T, which is essential for formation of the anammoxosome, which allowed reconstitution of the filaments for single particle cryo-EM work. While the resolution of the cryo-EM reconstruction is modest, combining this structure with the crystal structures of NRX-ABC and an NRX-T homolog provided insight into the assembly mechanisms.

Overall, the paper is well written and clear, and reveals the structure of a fascinating

multienzyme assembly. It also raises interesting questions about the mechanisms of nitrate generation. The structural work is well done, and the conclusions well supported by the structures.

I have only a few minor comments with respect to the helical reconstruction:

1) was the two-fold dyad symmetry, perpendicular to the helical axis, enforced during structure refinement? This isn't mentioned in the methods. If so this should be mentioned, if not it really ought to be as this would very likely improve the resolution of the structure.

We thank the reviewer for this valuable suggestion. Attempts have been made to refine the structure with a C2 symmetry. However, the resolution unfortunately didn't get any better and since we have a high-resolution crystal structure of NXR-ABC trimers, we do not expect any further insight into biological questions that could result from the improvement of the resolution of the EM maps.

2) the overlap between adjacent particles extracted from helices should be stated

We apologize that this was not sufficiently specified. The overlap is a third of the box size. The text has been changed accordingly.

3) what were the final refined helical symmetry parameters (rise/twist per subunit) for this reconstruction? Did these differ from the initial estimate from the subvolume average structure?

As the initial parameters for the helical reconstruction, we used fixed values obtained from a symmetry scan of the subtomogram average structure (twist: 110 degrees; rise: 114 Å). Setting the values to defined numbers was not successful even after several processing rounds. Our guess was that these were not absolute values (due to the low resolution of the StA structure) and we started a local search for symmetry with a range of 100-130 degrees for the twist and 100-130 Å for the rise. Moving the structure within a certain area was successful, which we assumed due to a certain degree of flexibility within the structure. However, Relion did not give the exact values at the end of the refinement, as these were not documented in the star file. The values for the twist of 121 degrees and for the rise of 108 Å have now been determined using Chimera, which is in the range of the symmetry scan of the subtomogram average structure. The values were added to Extended Data Table 3.

4) are there direct contacts between NXR-ABC trimers along the length of the helical axis? If so, these should be described, as well as whether they resemble any of the crystal packing interfaces observed in the crystal asymmetric unit.

The referee is right; the dimers-of-heterotrimers that, together with NXR-T, are the building blocks of the tubule structure are the same (within the limits of the EM resolution) as those observed in the asymmetric unit of the crystal structure. In the

crystal structure, to additional NXR-ABC heterotrimers takes the positions of the NXR-T subunits on either side of these dimers in the tubules. We now briefly mention this in the main text and discuss it in more detail in the supplementary information – this discussion was present in earlier versions but had fallen prey to an attempt to streamline the paper.

Decision Letter, first revision:

Dear Thomas,

Thank you for submitting your revised manuscript "Architecture of the Intracellular Superstructure-Forming Nitrite Oxidoreductase" (NMICROBIOL-21010091A) and for your patience as we waited to hear back from the reviewers. The paper has now been seen by the original referees and their comments are below. As you will see, the reviewers find that the paper has improved in revision and raise no additional points that need to be addressed from their part. Therefore we'll be very happy in principle to publish the manuscript in Nature Microbiology, pending a few minor revisions needed to comply with our editorial and formatting guidelines.

We are now performing detailed checks on your paper and all related files, and will send you a checklist detailing our editorial and formatting requirements in about a week. Please do not upload the final materials and make any revisions until you receive this additional information from us.

Congratulations once again on putting together such a nice story and thank you for your interest in Nature Microbiology. In the meantime, please do not hesitate to contact me if you have any questions.

{REDACTED}

Reviewer #1 (Remarks to the Author):

The authors have done an excellent job of responding to the suggested changes.

Reviewer #2 (Remarks to the Author):

The authors have adequately addressed my comments and requests during revision and it was a pleasure to read the revised manuscript. This study is a substantial contribution to both nitrogen cycle microbiology and structural biology. I have no further comments and congratulate the authors for their achievement!

Reviewer #3 (Remarks to the Author):

The authors have made the necessary changes and fully met my minor concerns.

Decision Letter, final checks:

Dear Thomas,

Thank you for your patience as we've prepared the guidelines for final submission of your Nature Microbiology manuscript, "Architecture of the Intracellular Superstructure-Forming Nitrite Oxidoreductase" (NMICROBIOL-21010091A). Please carefully follow the step-by-step instructions provided in the attached file, and add a response in each row of the table to indicate the changes that you have made. Please also check the attached Word document with suggested edits to the title and abstract. Ensuring that each point is addressed will help to ensure that your revised manuscript can be swiftly handed over to our production team.

In recognition of the time and expertise our reviewers provide to Nature Microbiology's editorial process, we would like to formally acknowledge their contribution to the external peer review of your manuscript entitled "Architecture of the Intracellular Superstructure-Forming Nitrite Oxidoreductase". For those reviewers who give their assent, we will be publishing their names alongside the published article.

Nature Microbiology offers a Transparent Peer Review option for new original research manuscripts submitted after December 1st, 2019. As part of this initiative, we encourage our authors to support increased transparency into the peer review process by agreeing to have the reviewer comments, author rebuttal letters, and editorial decision letters published as a Supplementary item. When you submit your final files please clearly state in your cover letter whether or not you would like to participate in this initiative. Please note that failure to state your preference will result in delays in accepting your manuscript for publication.

Cover suggestions

As you prepare your final files we encourage you to consider whether you have any images or illustrations that may be appropriate for use on the cover of Nature Microbiology. Covers should be both aesthetically appealing and scientifically relevant, and should be supplied at the best quality available. Due to the prominence of these images, we do not generally select images featuring faces, children, text, graphs, schematic drawings, or collages on our covers. We accept TIFF, JPEG, PNG or PSD file formats (a layered PSD file would be ideal), and the image should be at least 300ppi resolution (preferably 600-1200 ppi), in CMYK colour mode. If your image is selected, we may also use it on the journal website as a banner image, and may need to make artistic alterations to fit our journal style. Please submit your suggestions, clearly labeled, along with your final files. We'll be in touch if more information is needed.

Nature Microbiology has now transitioned to a unified Rights Collection system which will allow our Author Services team to quickly and easily collect the rights and permissions required to publish your work. Approximately 10 days after your paper is formally accepted, you will receive an email in providing you with a link to complete the grant of rights. If your paper is eligible for Open Access, our Author Services team will also be in touch regarding any additional information that may be required to arrange payment for your article. Please note that you will not receive your proofs until the publishing agreement has been received through our system.

Please note that *Nature Microbiology* is a Transformative Journal (TJ). Authors may publish their research with us through the traditional subscription access route or make their paper immediately open access through payment of an article-processing charge (APC). Authors will not be required to make a final decision about access to their article until it has been accepted. [Find out more about Transformative Journals](https://www.springernature.com/gp/open-research/transformative-journals)

Authors may need to take specific actions to achieve compliance with funder and institutional open access mandates. For submissions from January 2021, if your research is supported by a funder that requires immediate open access (e.g. according to [Plan S principles](https://www.springernature.com/gp/open-research/plan-s-compliance)) then you should select the gold OA route, and we will direct you to the compliant route where possible. For authors selecting the subscription publication route our standard licensing terms will need to be accepted, including our [self-archiving policies](https://www.springernature.com/gp/open-research/policies/journal-policies). Those standard licensing terms will supersede any other terms that the author or any third party may assert apply to any version of the manuscript.

When you are ready, please use the following link for uploading all the required materials:

{REDACTED}

In the meantime, please do not hesitate to contact us if you have any questions.

{REDACTED}

Final Decision Letter:

Dear Thomas,

I am very pleased to accept your Article "Structural and functional characterization of the intracellular filament-forming Nitrite Oxidoreductase multiprotein complex" for publication in Nature Microbiology.

Thank you for having chosen to submit your work to us and many congratulations to you and your co-authors.

Before your manuscript is typeset, we will edit the text to ensure it is intelligible to our wide readership and conforms to house style. We look particularly carefully at the titles of all papers to ensure that they are relatively brief and understandable.

Acceptance of your manuscript is conditional on all authors' agreement with our publication policies (see www.nature.com/nmicrobiolate/authors/gta/content-type/index.html). In particular your manuscript must not be published elsewhere and there must be no announcement of the work to any media outlet until the publication date (the day on which it is uploaded onto our website).

Please note that *Nature Microbiology* is a Transformative Journal (TJ). Authors may publish their research with us through the traditional subscription access route or make their paper immediately open access through payment of an article-processing charge (APC). Authors will not be required to make a final decision about access to their article until it has been accepted. [Find out more about Transformative Journals](https://www.springernature.com/gp/open-research/transformative-journals)

Authors may need to take specific actions to achieve compliance with funder and institutional open access mandates. For submissions from January 2021, if your research is supported by a funder that requires immediate open access (e.g. according to [Plan S principles](https://www.springernature.com/gp/open-research/plan-s-compliance)) then you should select the gold OA route, and we will direct you to the compliant route where possible. For authors selecting the subscription publication route our standard licensing terms will need to be accepted, including our [self-archiving policies](https://www.springernature.com/gp/open-research/policies/journal-policies). Those standard licensing terms will supersede any other terms that the author or any third party may assert apply to any version of the manuscript.

An online order form for reprints of your paper is available at <https://www.nature.com/reprints/author->

reprints.html"><https://www.nature.com/reprints/author-reprints.html>. All co-authors, authors' institutions and authors' funding agencies can order reprints using the form appropriate to their geographical region.

Congratulations once again to you and your colleagues for putting together such a nice story, I look forward to seeing it published